# A *cis*-carotene derived apocarotenoid regulates etioplast and chloroplast development

Christopher I Cazzonelli[1†]*, Xin Hou[2†], Yagiz Alagoz[1], John Rivers[2], Namraj Dhami[1], Jiwon Lee[3], Shashikanth Marri[2], Barry J Pogson[2]*

[1]Hawkesbury Institute for the Environment, Western Sydney University, Penrith, Australia; [2]Research School of Biology, The Australian National University, Canberra, Australia; [3]Centre for Advanced Microscopy, The Australian National University, Canberra, Australia

**Abstract** Carotenoids are a core plastid component and yet their regulatory function during plastid biogenesis remains enigmatic. A unique carotenoid biosynthesis mutant, *carotenoid chloroplast regulation 2* (*ccr2*), that has no prolamellar body (PLB) and normal PROTOCHLOROPHYLLIDE OXIDOREDUCTASE (POR) levels, was used to demonstrate a regulatory function for carotenoids and their derivatives under varied dark-light regimes. A forward genetics approach revealed how an epistatic interaction between a *ζ-carotene isomerase* mutant (*ziso-155*) and *ccr2* blocked the biosynthesis of specific *cis*-carotenes and restored PLB formation in etioplasts. We attributed this to a novel apocarotenoid retrograde signal, as chemical inhibition of carotenoid cleavage dioxygenase activity restored PLB formation in *ccr2* etioplasts during skotomorphogenesis. The apocarotenoid acted in parallel to the repressor of photomorphogenesis, DEETIOLATED1 (DET1), to transcriptionally regulate PROTOCHLOROPHYLLIDE OXIDOREDUCTASE (POR), PHYTOCHROME INTERACTING FACTOR3 (PIF3) and ELONGATED HYPOCOTYL5 (HY5). The unknown apocarotenoid signal restored POR protein levels and PLB formation in *det1*, thereby controlling plastid development.

*For correspondence:
c.cazzonelli@westernsydney.edu.au (CIC);
barry.pogson@anu.edu.au (BJP)

†These authors contributed equally to this work

Competing interests: The authors declare that no competing interests exist.

## Introduction

Carotenoids are a diverse group of hydrophobic isoprenoid pigments required for numerous biological processes in photosynthetic organisms and are essential for human health (*Cazzonelli, 2011*; *Baranski and Cazzonelli, 2016*). In addition to providing plant flowers, fruits and seeds with distinct colours, carotenoids have accessory roles in facilitating the assembly of the light harvesting complex, light capture during photosynthesis and photoprotection during high light and/or temperature stress (*Nisar et al., 2015*; *Baranski and Cazzonelli, 2016*). The current frontiers are to discover the regulators of carotenoid biosynthesis, storage, and catabolism and apocarotenoids that in turn regulate plant development and photosynthesis (*Cazzonelli and Pogson, 2010*; *Havaux, 2014*; *Baranski and Cazzonelli, 2016*; *Hou et al., 2016*).

In higher plants, *cis*-carotene biosynthesis is initiated by the condensation of two molecules of geranylgeranyl diphosphate (GGPP) to form phytoene, which is catalyzed by the rate-limiting enzyme phytoene synthase (PSY) (*Von Lintig et al., 1997*; *Li et al., 2008*; *Rodríguez-Villalón et al., 2009*; *Welsch et al., 2010*; *Zhou et al., 2015*) (*Figure 1—figure supplement 1A*). Next, phytoene desaturase (PDS), ζ-carotene desaturases (ZDS), ζ-carotene isomerase (ZISO) and *cis-trans*-carotene isomerase (CRTISO) convert the colourless phytoene into the pinkish-red coloured all-*trans*-lycopene (*Bartley et al., 1999*; *Isaacson et al., 2002*; *Park et al., 2002*; *Dong et al., 2007*; *Chen et al., 2010*; *Yu et al., 2011*). In the dark, the isomerisation of tri-*cis*-ζ-carotene to di-*cis*-ζ-carotene and

tetra-*cis*-lycopene to all-*trans*-lycopene has a strict requirement for ZISO and CRTISO activity respectively (*Park et al., 2002*; *Chen et al., 2010*). However, light-mediated photoisomerisation in the presence of a photosensitiser can substitute for a lack of isomerase activity (*Giuliano et al., 2002*; *Vijayalakshmi et al., 2015*; *Alagoz et al., 2018*).

The carotenoid biosynthetic pathway branches after lycopene to produce α/β-carotenes (*Figure 1—figure supplement 1A*) (*Cunningham et al., 1993*; *Cunningham et al., 1996*; *Pecker et al., 1996*; *Ronen et al., 1999*). α-carotene and β-carotene are further hydroxylated to produce the oxygenated carotenoids called xanthophylls (e.g. lutein, violaxanthin and zeaxanthin), which comprise the most abundant carotenoids found in photosynthetic leaves. Carotenoids are precursors for apocarotenoids (carotenoid cleavage products) such as phytohormones abscisic acid (ABA) and strigolactone (SL) as well as other apocarotenoids that function in root-mycorrhizal interactions, leaf development, acclimation to environmental stress and retrograde signalling (*Havaux, 2014*; *Walter et al., 2015*; *Chan et al., 2016*; *Hou et al., 2016*). The carotenoid cleavage dioxygenase and nine-*cis*-epoxy-carotenoid dioxygenase (CCD/NCED) family cleave carotenoids to yield apocarotenoids (*Hou et al., 2016*). The CCDs have substrate preferences depending on the tissue and nature of the assay (*Walter and Strack, 2011*; *Harrison and Bugg, 2014*; *Bruno et al., 2016*). The five members of the NCED sub-group are exclusively involved in cleavage of violaxanthin and neoxanthin to form ABA (*Finkelstein, 2013*). The four CCDs have well defined roles in carotenoid degradation in seeds (CCD1 and CCD4) and the synthesis of strigolactones (CCD7/MAX3 and CCD8/MAX4) (*Auldridge et al., 2006*; *Gonzalez-Jorge et al., 2013*; *Ilg et al., 2014*; *Al-Babili and Bouwmeester, 2015*). Non-enzymatic oxidative cleavage of carotenoids can also generate apocarotenoids by singlet oxygen ($^1O_2$)-mediated photo-oxidation or by lipoxygenase and peroxidase-mediated co-oxidation (*Leenhardt et al., 2006*; *González-Pérez et al., 2011*). Non-enzymatic carotenoid degradation acts preferentially on selective molecules such as β-carotene and its apocarotenoid derivatives.

*cis*-carotenes such as phytoene, phytofluene and tetra-*cis*-lycopene are reported to be resistant to non-enzymatic degradation (*Schaub et al., 2018*), although there are some reports that CCDs cleave specific *cis*-carotenes in vitro (*Bruno et al., 2016*). Whether there is a physiological relevance for a *cis*-carotene derived cleavage product or apocarotenoid signal (ACS) in vivo, remains unclear. A hunt is on to identify a *cis*-carotene cleavage product that functions as a retrograde signal to fine-tune nuclear gene expression during development or in response to stress (*Kachanovsky et al., 2012*; *Fantini et al., 2013*; *Avendaño-Vázquez et al., 2014*; *Álvarez et al., 2016*). CCD4 was implicated in the generation of a *cis*-carotene-derived apocarotenoid signal that regulates leaf shape, chloroplast and nuclear gene expression in the Arabidopsis *clb5/zds* (chloroplast biogenesis-5 / ζ-carotene desaturase) mutant (*Avendaño-Vázquez et al., 2014*). A metabolon regulatory loop around all-*trans*-ζ-carotene was proposed in tomato fruit that can sense *cis*-carotene accumulation, their derivatives or the enzymes themselves (*Fantini et al., 2013*). The accumulation of *cis*-carotenes in tomato fruit have also been linked to the metabolic feedback-regulation of *PSY* transcription and translation (*Kachanovsky et al., 2012*; *Álvarez et al., 2016*). Therefore, *cis*-carotenes themselves or their cleavage products appear to have functional roles, of which the targets and regulatory mechanism(s) remains unknown.

Determining a mechanistic function for *cis*-carotenes in planta has been challenged by low levels of *cis*-carotene accumulation in wild type tissues. When the upper carotenoid pathway is perturbed (*Figure 1—figure supplement 1A*) (*Alagoz et al., 2018*), seedling lethality (*psy*, *pds* and *zds*), impaired chlorophyll and *cis*-carotene accumulation (*ziso* and *crtiso*) as well as a reduction in lutein (*crtiso*) become apparent (*Isaacson et al., 2002*; *Park et al., 2002*). *ziso* mutants in maize (*y9*) and Arabidopsis (*zic*) display transverse pale-green zebra-striping patterns and delayed cotyledon greening respectively, resembling impaired chloroplast development that causes a leaf virescence phenotype (*Janick-Buckner et al., 2001*; *Li et al., 2007*; *Chen et al., 2010*). Similarly, *crtiso* loss-of-function in tomato (*tangerine*), melon (*yofi*) and rice (*zebra*) mutants show varying degrees of unexplained leaf virescence (*Isaacson et al., 2002*; *Park et al., 2002*; *Chai et al., 2011*; *Galpaz et al., 2013*), of which the cause triggering this phenomena could be attributed to light/dark cycles even though carotenoid composition remained unaffected (*Han et al., 2012*).

During skotomorphogenesis prolamellar bodies (PLB) develop in etioplasts of seedling tissues. The PLB is a crystalline agglomeration of protochlorophyllide (PChlide), POR enzyme and fragments of pro-thylakoid membranes. The PLB provides a structural framework for the light-catalysed

conversion of PChlide into chlorophylls by POR within picoseconds in conjunction with the assembly of the photosynthetic apparatus (*Sundqvist and Dahlin, 1997*; *Sytina et al., 2008*). The de-etiolation of seedlings upon exposure to light activates a sophisticated network consisting of receptors, genetic and biochemical signals that trigger photomorphogenesis. Changes in light-induced morphogenesis include: short hypocotyls; expanded and photosynthetically-active cotyledons with developing chloroplasts; and activation of self-regulated stem cell populations at shoot apices (*Arsovski et al., 2012*; *Lau and Deng, 2012*). DETIOLATED1 (DET1) and CONSTITUTIVE PHOTO-MORPHOGENIC 1 (COP1) promote skotomorphogenesis, while *det1* and *cop1* mutants lack POR and cannot assemble a PLB. They broadly promote photomorphogenesis in the dark (*Chory et al., 1989*; *Sperling et al., 1998*; *Datta et al., 2006*) (*Figure 1—figure supplement 1B*). This is a consequence of DET1 and COP1 post-transcriptionally controlling the levels of PHYTOCHROME INTER-ACTING FACTOR 3 (PIF3; constitutive transcriptional repressor of photomorphogenesis) and ELONGATED HYPOCOTYL 5 (HY5; positive transcriptional regulator of photomorphogenesis) that control PORA and *PHOTOSYNTHESIS ASSOCIATED NUCLEAR GENE* (*PhANG*) expression (*Stephenson et al., 2009*; *Lau and Deng, 2012*; *Xu et al., 2016*; *Llorente et al., 2017*). Thus, in the dark, wild-type plants accumulate PIF3, but lack HY5, conversely *det1* lacks PIF3 and accumulates HY5 protein (*Figure 1—figure supplement 1B*).

PLB formation occurs in carotenoid deficient mutants. Norflurazon (NF) treated wheat seedlings grown in darkness lack carotenoids, other than phytoene (*Figure 1—figure supplement 1A*), and yet still form a PLB that is somewhat aberrant in having a looser attachment of POR to the lipid phase and which dissociates early from the membranes during photomorphogenesis (*Denev et al., 2005*). In contrast, *ccr2* is similar to *cop1/det1* mutants in that it lacks a PLB in etioplasts, yet it is unique among PLB-deficient mutants in having normal PChlide and POR protein levels (*Park et al., 2002*). The associated hyper accumulation of *cis*-carotenes led to the untested hypothesis that *cis*-carotenes structurally prevent PLB formation in etioplasts of dark germinated *ccr2* during skotomorphogenesis and this in turn delayed cotyledon greening following illumination (*Park et al., 2002*; *Datta et al., 2006*; *Cuttriss et al., 2007*). However, it was never apparent why other carotenes, such as 15-*cis*-phytoene and all-*trans*-lycopene, permitted PLB formation, and whether there were regulatory functions for the *cis*-carotenes themselves, or their cleavage products that accumulate in *ccr2*.

In this paper, we describe how changes in photoperiod are sufficient to perturb or permit plastid development in *ccr2*, the former leading to leaf virescence. A revertant screen of *ccr2* revealed new connections between a *cis*-carotene-derived signalling metabolite, PLB formation during skotomorphogenesis and chloroplast development following photomorphogenesis. We demonstrate how an unidentified apocarotenoid signal acted in parallel to DET1 to control PLB formation as well as *POR*, *PIF3* and *HY5* transcript levels, thereby fine-tuning plastid development in tissues exposed to extended periods of darkness.

## Results

### A shorter photoperiod perturbs chloroplast biogenesis and promotes leaf virescence

The *crtiso* mutants have been reported to display different leaf pigmentation phenotypes in a species-independent manner, with rice and tomato showing yellow and green sectors resembling signs of virescence, but no such observations have been made in Arabidopsis. To address the species-dependence we investigated if light regimes affected leaf pigment levels and hence plastid development in Arabidopsis *crtiso* (*ccr2*) mutants. Growing *ccr2* plants at a lower light intensity of 50 µE during a long 16 hr photoperiod did not cause any obvious changes in morphology or leaf virescence (*Figure 1—figure supplement 2A*). In contrast, an 8 hr photoperiod resulted in newly emerged *ccr2* leaves to appear yellow in pigmentation (*Figure 1—figure supplement 2B*) due to a substantial reduction in total chlorophyll (*Figure 1—figure supplement 2D*). As development progressed the yellow leaf (YL) phenotype became less obvious and greener leaves (GL) developed (*Figure 1—figure supplement 2C*). Therefore, by reducing the photoperiod we were able to replicate the leaf virescence phenotype in Arabidopsis previous reported to occur in tomato and rice (*Isaacson et al., 2002*; *Chai et al., 2011*). The manifestation of virescence in both *zebra2* (*Han et al., 2012*) and *ccr2*

(*Figure 1—figure supplement 2B*) mutant leaves grown under a shorter photoperiod revealed that the phenotype was dependent upon an extended period of darkness.

Next, we demonstrated that day length affects plastid development in newly emerged leaf tissues undergoing cellular differentiation. We replicated the YL phenotype by shifting three weeks old *ccr2* plants from a long 16 hr to shorter 8 hr photoperiod (*Figure 1A–B*). The newly emerged leaves of *ccr2* appeared yellow and virescent, while leaves that developed under a 16 hr photoperiod remained green similar to wild type (*Figure 1B*). Consistent with the phenotype, the yellow sectors of *ccr2* displayed a 2.4-fold reduction in total chlorophyll levels, while mature green leaf sectors formed prior to the photoperiod shift had chlorophyll levels similar to that of WT (*Figure 1C*). The chlorophyll *a/b* as well as carotenoid/chlorophyll ratios were not significantly different (*Figure 1C*). Consistent with the reduction in chlorophyll, total carotenoid content in yellow sectors of *ccr2* was reduced due to lower levels of lutein, β-carotene and neoxanthin (*Figure 1D*). The percentage composition of zeaxanthin and antheraxanthin was significantly enhanced in yellow sectors, perhaps reflecting a greater demand for xanthophyll cycle pigments that reduce photo-oxidative damage (*Figure 1—figure supplement 2E*). Transmission electron microscopy (TEM) revealed that yellow *ccr2* leaf sectors contained poorly differentiated chloroplasts lacking membrane structures consisting of thylakoid and grana stacks, as well as appearing spherical in shape, rather than oval when compared to green leaf tissues from WT or *ccr2* (*Figure 1E*). Therefore, chloroplast development can be perturbed in *ccr2* leaf primordia cells that develop under extended periods of darkness leading to changes in pigment content.

## The leaf virescence phenotype correlated with *cis*-carotene accumulation

We next investigated the relationship between photoperiod, perturbations in carotenogenesis and plastid development. Green leaf tissues from *ccr2* have an altered proportion of β-xanthophylls at the expense of less lutein, yet plants grown under a longer photoperiod show normal plastid development (*Park et al., 2002*). Reducing the photoperiod could limit the photoisomerisation of tetra-*cis*-lycopene to all-*trans*-lycopene and alter ABA and/or strigolactone biosynthesis (*Supplementary file 1*). Therefore, *ccr2*, *lycopene epsilon cyclase* (*lut2*; *lutein deficient 2*), *zeaxanthin epoxidase* (*aba1-3*; *aba deficient 1*) and *carotenoid cleavage dioxygenase 8* (*max 4*; *more axillary branching 4*) mutants were shifted from a 16 hr to 8 hr photoperiod (*Figure 2A*). *ccr2* showed a clear virescent yellow leaf phenotype, while the other mutants produced green leaves similar to those of WT. Therefore, we could not attribute yellow leaf virescence to a reduction in lutein, perturbation of SL or ABA biosynthesis.

Next, we tested if the *ccr2* yellow leaf phenotype was linked to the accumulation of *cis*-carotenes in the pathway upstream of all-*trans*-lycopene. Mutations in *PSY*, *PDS* and *ZDS* cause leaf bleaching and are not viable in soil. Alternatively, *carotenoid chloroplast regulator 1* (*ccr1* or otherwise known as *sdg8*; *set domain group 8*) and *ζ-carotene isomerase* (*ziso*) mutants are viable and accumulate *cis*-carotenes in etiolated tissues (*Cazzonelli et al., 2009b*; *Chen et al., 2010*). Indeed, both *ccr1* and *ziso* displayed a partial yellow leaf phenotype near the zone of cellular differentiation (e.g. petiole-leaf margin), however unlike *ccr2* the maturing leaf tissues greened rapidly such that *ziso* was more similar to WT than *ccr2* (*Figure 2A*).

Does a shorter photoperiod lead to the accumulation of *cis*-carotenes in newly emerged leaf tissues of *ccr2* displaying altered plastid development? First, we tested if an extended dark period (6 days) would result in the accumulation of *cis*-carotenoids in mature (3 weeks) rosette leaf tissues. Compared to adult WT leaves, prolonged darkness resulted in notable yellowing of *ccr2* leaves and clearly discernible accumulation of tetra-*cis*-lycopene, neurosporene isomers, ζ-carotene, phytofluene and phytoene (*Figure 2B*). We next shifted three-week-old plants from a 16 hr to 8 hr photoperiod and the yellow sectors from newly emerged *ccr2* leaves accumulated detectable levels of *cis*-lycopene, neurosporene isomers, ζ-carotene, phytofluene and phytoene (*Figure 2C*). Interestingly, even when plants were grown under a 16 hr photoperiod, we could detect phytofluene and phytoene in floral buds as well as newly emerged rosette leaves from *ccr2*, and at trace levels in WT (*Figure 2D*). In addition, a higher ratio of phytofluene and phytoene relative to β-carotene was observed in newly emerged *ccr2* tissues, which coincided with a lower percentage of lutein when compared to older tissues.

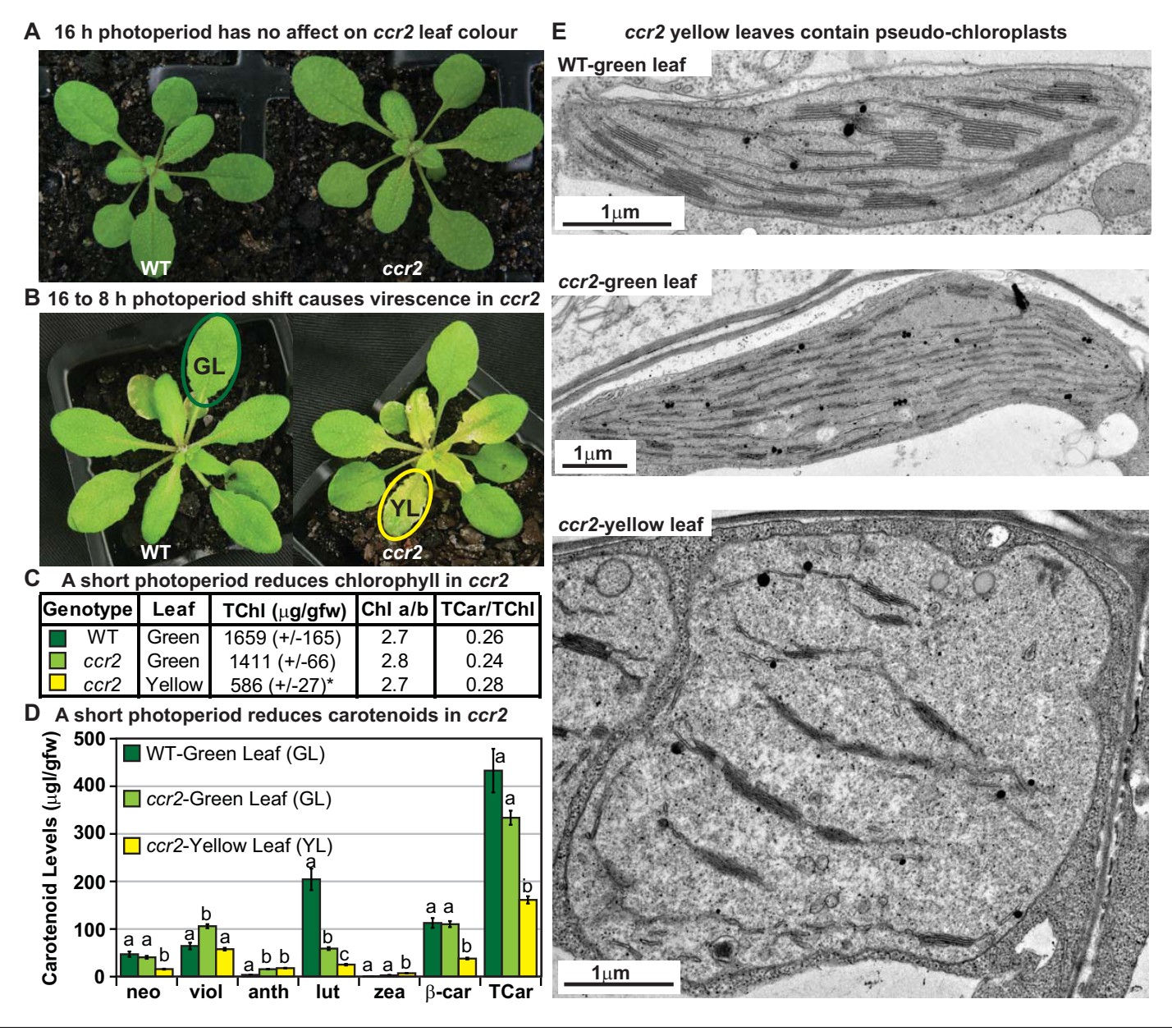

**Figure 1.** A shorter photoperiod alters plastid development and pigmentation in *ccr2*. (A) Three-week-old wild type (WT) and *ccr2* plants growing under a 16 hr light photoperiod. (B) Two-week-old plants were shifted from a 16 hr to 8 hr photoperiod for one week and newly emerged or expanded leaves appeared yellow in *ccr2* (YL; yellow outline), while WT displayed green leaves (GL; green outline). (C) Chlorophyll levels (μg/gfw) and pigment ratios in green (WT and *ccr2*) and yellow (*ccr2*) leaves formed one week after a photoperiod shift from 16 hr to 8 hr. Standard error is shown for TChl (n = 5, single leaf from five plants). Star denotes significant differences (ANOVA; $p < 0.05$). (D) Absolute carotenoid levels (μg/gfw) in green (WT and *ccr2*) and yellow (*ccr2*) leaves formed one week after a photoperiod light shift from 16 hr to 8 hr. Values represent average and standard error bars are displayed (n = 5, single leaf from five plants). Lettering denotes significance (ANOVA; $p < 0.05$). Neoxanthin (neo), violaxanthin (viol), antheraxanthin (anth), lutein (lut), zeaxanthin (zea), β-carotene (β-car), Total Chlorophyll (TChl), Chlorophyll a/b ratio (Chl a/b), Total carotenoids (TCar). (E)Transmission electron micrograph images showing representative chloroplasts from WT and *ccr2* green leaf sectors as well as yellow leaf sectors of *ccr2*.
The online version of this article includes the following figure supplement(s) for figure 1:

**Figure supplement 1.** *cis*-carotene biosynthesis and regulation of PLB formation during skotomorphogenesis.
**Figure supplement 2.** A shorter photoperiod promotes yellow leaf virescence affecting chlorophyll levels and carotenoid composition in *ccr2*.

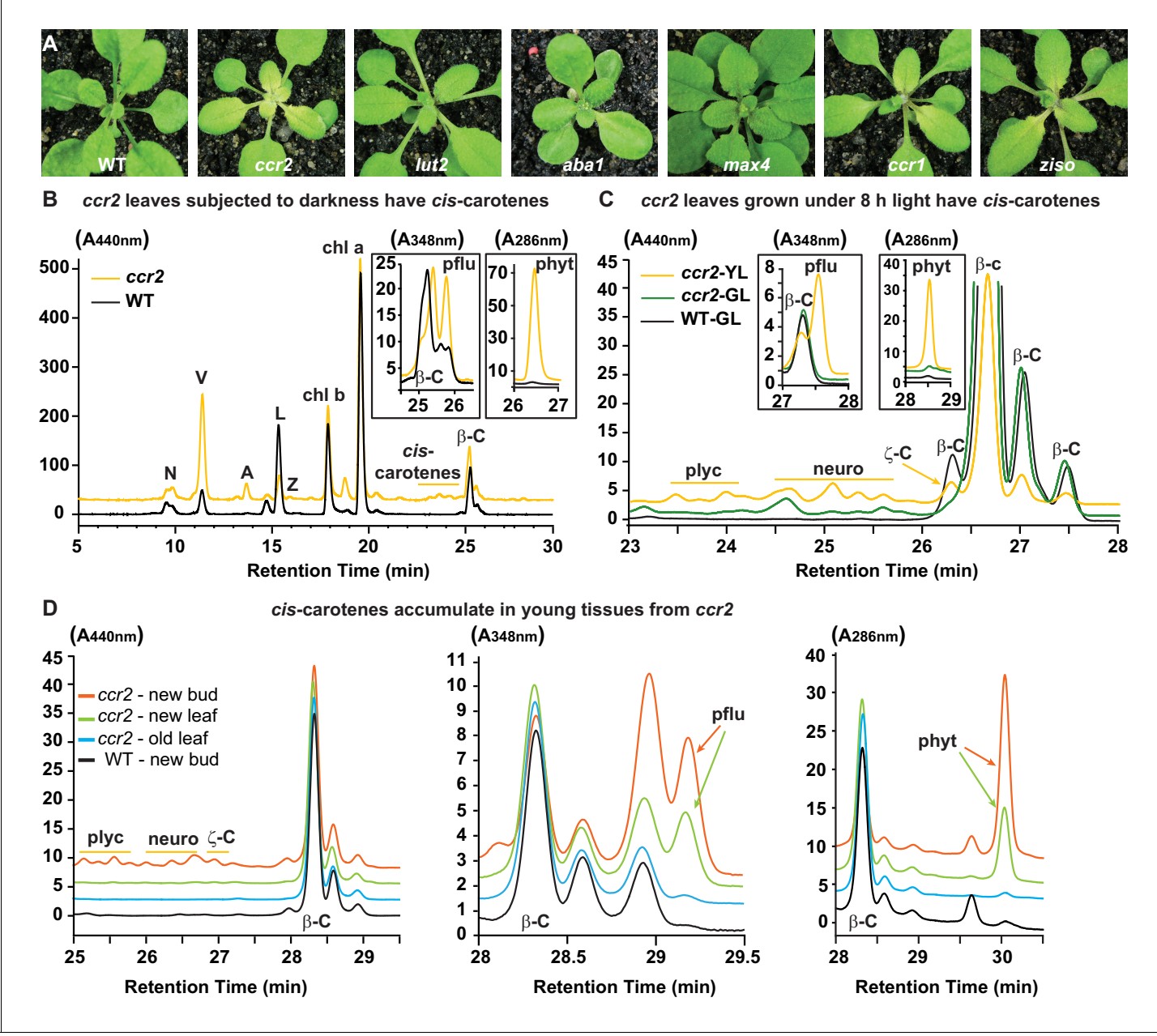

**Figure 2.** Altered plastid development in *ccr2* is linked with *cis*-carotene accumulation and not to a perturbation in ABA or SL. (**A**) Mutants that perturb the levels of lutein, ABA, SL and accumulate *cis*-carotenes (*ccr2*, *ccr1* and *ziso*) were grown for two weeks under a 16 hr photoperiod and then shifted to a shorter 8 hr photoperiod for one week. Representative images showing newly emerged and expanding leaves from multiple experimental and biological repetitions (n > 20 plants per line) are displayed. Genetic alleles tested include Col-0 (WT), *ccr2-1* (*carotenoid isomerase*), *lut2-1* (*epsilon lycopene cyclase*), *aba1-3* (Ler background) (*zeaxanthin epoxidase*), *max4/ccd8* (carotenoid cleavage dioxygenase 8), *ccr1-1/sdg8* (*set domain group 8*) and *ziso1-3* (*ζ-carotene isomerase*). (**B**) Carotenoid profiles in rosette leaves from three-week-old plants grown under a 16 hr photoperiod and subjected to 6-d of extended darkness. (**C**) Carotenoid profiles in three-week-old rosette leaves from plants grown under a constant 8 hr light photoperiod. Pigments were profiled in a yellow leaf (YL) and green leaf (GL) from WT and *ccr2*. (**D**) Carotenoid profiles in newly emerged floral bud and rosette leaf tissues harvested from four-week-old plants growing under a 16 hr photoperiod. Carotenoid profile traces of various tissue extracts from wild type (WT) and *ccr2* show pigments at wavelengths close to the absorption maxima of $A_{440nm}$ (Neoxanthin; N, violaxanthin; V, antheraxanthin; A, lutein; L, zeaxanthin; Z, β-carotene isomers; β-C, chlorophyll a; Chl a, chlorophyll b; chl b, tetra-*cis*-lycopene; plyc, neurosporene isomers; neuro, and ζ-carotene; ζ-C), $A_{348nm}$ (phytofluene; pflu) and $A_{286nm}$ (phytoene; phyt). HPLC profile y-axis units are in milli-absorbance units (mAU). HPLC traces are representative of multiple leaves from multiple experimental repetitions and retention times vary due to using different columns.

## Second site genetic reversion restored plastid development in *ccr2*

We undertook a revertant screen to identify second site genes mutations in which proteins could complement the plastid development in *ccr2*, while still maintaining a perturbed carotenoid profile. Twenty-five revertant lines reproducibly displayed green immature leaves in response to a photoperiod shift, as exemplified by r*ccr2*⁻¹⁵⁴ and r*ccr2*⁻¹⁵⁵ (*Figure 3A*). Leaf tissues of all *rccr2* lines contained reduced lutein and xanthophyll composition similar to *ccr2* (*Figure 3B*). When grown under a shorter photoperiod, *rccr2* lines produced greener rosettes with less yellow virescence compared to *ccr2* and chlorophyll levels were similar to WT (*Figure 3C–D*).

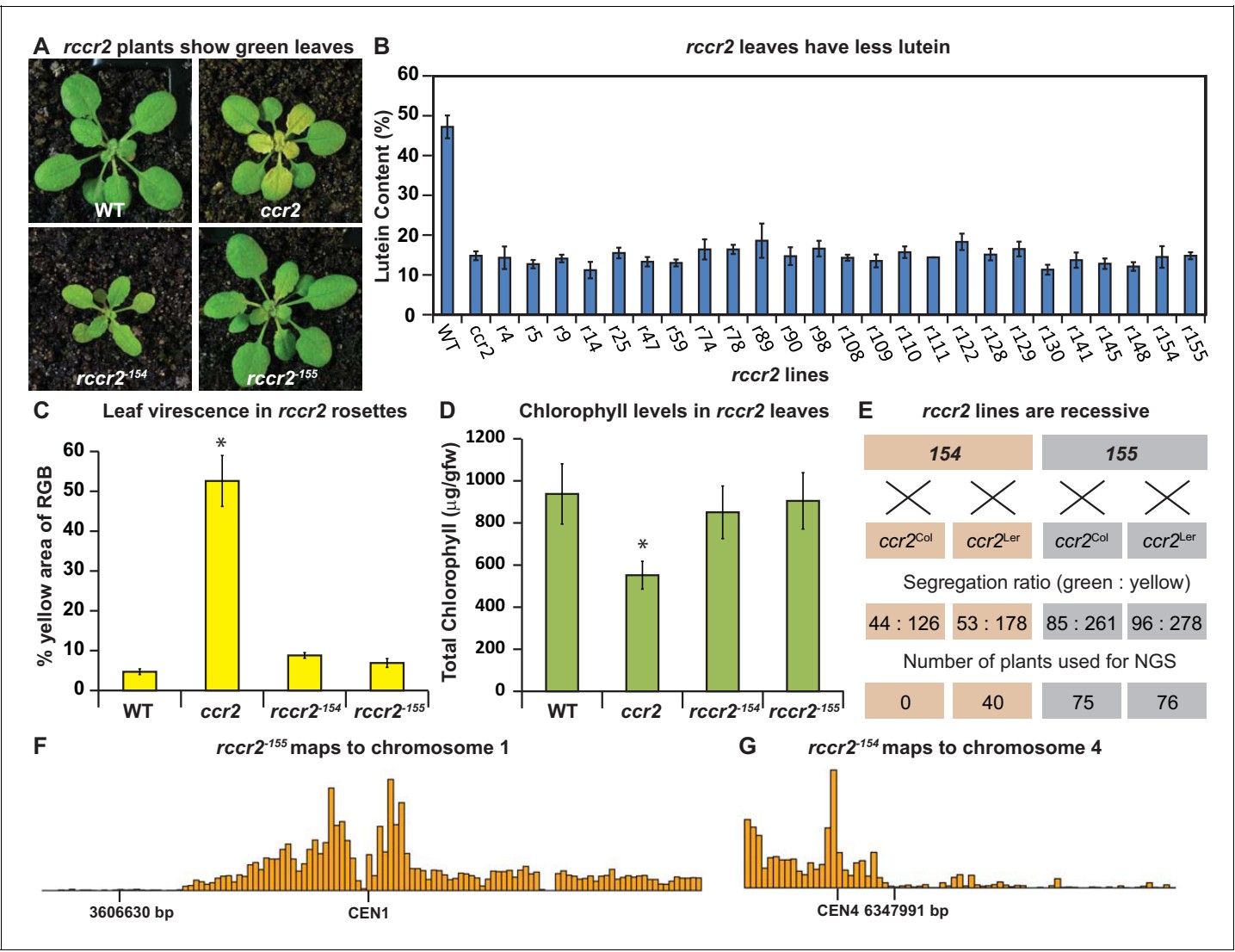

**Figure 3.** A forward genetics screen identified revertant lines of *ccr2* having reduced lutein and normal chlorophyll accumulation when grown under a shorter photoperiod. (A) Representative images of *rccr2*⁻¹⁵⁵ and *rccr2*⁻¹⁵⁴ rosettes one week after shifting two-week old plants from a 16 hr to 8 hr photoperiod. (B) Percentage lutein relative to total carotenoids in immature leaves from WT, *ccr2* and *rccr2* lines. (C) The degree of leaf virescence detected in rosettes following a reduction in photoperiod. Leaf virescence (% of yellow relative to RGB; Red-Green-Blue) in WT, *ccr2*, *rccr2*⁻¹⁵⁴ and *rccr2*⁻¹⁵⁵ rosettes was quantified using the Lemnatec Scanalyser system and software. (D) Total chlorophyll content in rosette leaves from WT, *ccr2*, *rccr2*⁻¹⁵⁴ and *rccr2*⁻¹⁵⁵ plants exposed to a shorter photoperiod. (E) Segregation ratios of *rccr2*⁻¹⁵⁴ and *rccr2*⁻¹⁵⁵ after backcrossing to the *ccr2* parent in both Columbia (Col-0) and Landsberg erecta (Ler) ecotypes. (NGS; next generation sequencing) (F) and (G). Leaves were pooled from a segregating F₂ progeny of *rccr2*⁻¹⁵⁵ (F) and *rccr2*⁻¹⁵⁴ (G) plants and genomic DNA purified for NGS. Bars reflect independent polymorphisms for Ler and/or Columbia SNPs across the Chromosome. The SNP desert indicates there is only Columbia SNP, indicating linkage disequilibrium and less recombination around the location of the causative mutation for *ziso*⁻¹⁵⁵ (3606630 bp; G to A) and *det1*⁻¹⁵⁴ (6347991 bp; G to A). Error bars denote standard error of means (SEM) and stars denote statistical significance (ANOVA; p<0.05).

In order to establish a segregating population for next generation mapping (NGM) *rccr2* lines were backcrossed to the original *ccr2* parent (Col-0) and/or a *ccr2* line established in the Landsberg erecta background (L*ccr2*). All *rccr2* lines were recessive for the reversion of shorter photoperiod dependent yellow leaves (e.g. r*ccr2*$^{-154}$ and r*ccr2*$^{-155}$; **Figure 3E**). Next generation sequencing (NGS) technologies were used to deep sequence the genomic DNA (gDNA) from leaves of homozygous (M$_2$) plants to identify non-recombinant deserts in chromosome 1 (3605576 bp) and chromosome 4 (6346463 bp) for both r*ccr2*$^{-155}$ and r*ccr2*$^{-154}$, respectively (**Figure 3F–G**). Both non-recombinant deserts contained SNPs displaying a discordant chastity value of approximately 1.0 representing the causal mutation of interest (**Austin et al., 2011**).

## An epistatic interaction between ziso and ccr2 revealed specific *cis*-carotenes perturb PLB formation

r*ccr2*$^{-155}$ lacked recombination at the bottom arm of chromosome one surrounding a single nucleotide polymorphism (G-A mutation at 3606630 bp) within exon 3 of the *ZISO* gene (639 bp of mRNA), hereafter referred as *ccr2 ziso-155* (**Figure 4A**). This polymorphism caused a premature stop codon leading to a truncated ZISO protein (212 instead of 367 amino acids). The overexpression of the functional *ZISO* cDNA fragment in *ccr2 ziso-155* restored *ccr2* leaf yellowing in plants grown under an 8 hr photoperiod (**Figure 4B**). A double mutant generated by crossing *ccr2* with *ziso1-4* further confirmed the loss-of-function in *ziso* can restore plastid development in newly emerged immature leaves of *ccr2*. Carotenoid analysis of immature leaf tissues of *ccr2 ziso-155* revealed reduced lutein and xanthophyll composition similar to *ccr2*, indicating that the complementation of the YL was not due to a change in xanthophyll levels (**Figure 3B**). The epistatic nature between *ziso* and *crtiso* revealed that a specific *cis*-carotene downstream of *ZISO* activity perturbed plastid development.

Analysis of the *cis*-carotene profile in etiolated cotyledons showed that *ccr2 ziso1-4* had an identical carotenoid profile to that of *ziso* in that it could only accumulate 9,15,9'-tri-*cis*-ζ-carotene, phytofluene and phytoene (**Figure 4C**). In contrast, *ccr2* accumulated lower levels of these three compounds, yet higher quantities of 9, 9'-di-*cis* ζ-carotene, 7,9,9'-tri-*cis*-neurosporene and 7,9,9',7'-tetra-*cis*-lycopene, all of which were undetectable in the *ziso* background (**Figure 4C**, **Table 1**). Therefore, *ziso* blocks the biosynthesis of neurosporene isomers, tetra-*cis*-lycopene and di-*cis*-ζ-carotene under shorter photoperiods, and they themselves or their cleavage products appear to disrupt plastid development in *ccr2*.

How are the specific *cis*-carotenes disrupting plastid development? We first examined etiolated cotyledons of WT, *ccr2*, *ziso1-4* and *ccr2 ziso-155*. We confirmed *ccr2* lacked a PLB in all sections examined (**Figure 4D**, **Supplementary file 2**). We observed 66% of *ziso1-4* etioplasts contained PLBs (**Figure 4D**, **Supplementary file 2**). Intriguingly, the vast majority (>94%) of etioplasts examined from *ccr2 ziso-155* and *ccr2 ziso1-4* contained a PLB (**Figure 4D**, **Supplementary file 2**). Cotyledon greening of de-etiolated seedlings revealed a significant delay in chlorophyll accumulation for both *ccr2* and *ziso1-4* when compared to WT after 24, 48 and 72 hr of continuous white light (**Figure 4E**). The reduced levels of chlorophyll in *ziso1-4* were not as severe as *ccr2*, consistent with *ziso1-4* showing a slight virescent phenotype in comparison to the strong one of *ccr2* (**Figure 2A**). Cotyledons of the *ccr2 ziso-155* and *ccr2 ziso1-4* double mutants accumulated levels of chlorophyll similar to that of WT, 48 and 72 hr following de-etiolation (**Figure 4E**). We conclude that a specific *cis*-carotene produced in *ccr2* prevents PLB formation during skotomorphogenesis and perturbs chloroplast development during de-etiolation.

## The activation of photosynthesis associated nuclear gene expression restored PLB formation in *ccr2*

The transcriptomes of WT, *ccr2* and *ccr2 ziso-155* etiolated seedlings (ES), yellow emerging juvenile leaves (JL) from *ccr2*, and green JL leaves from WT and *ccr2 ziso-155* were assessed using RNA sequencing analysis. Compared to WT there were 2- to 4-fold less differentially expressed (DE) genes in *ccr2* (ES;191 and JL;1217) than for *ccr2 ziso-155* (ES;385 and JL;5550). Gene ontology (GO) analysis revealed a DE gene list significantly enriched in metabolic processes and stress responses in both tissue types of *ccr2*. Etiolated tissues of *ccr2* showed DE genes enriched in photosynthetic processes (17/191; FDR < 3.8xE$^{-06}$) that were not apparent in *ccr2 ziso-155*, which had DE genes more

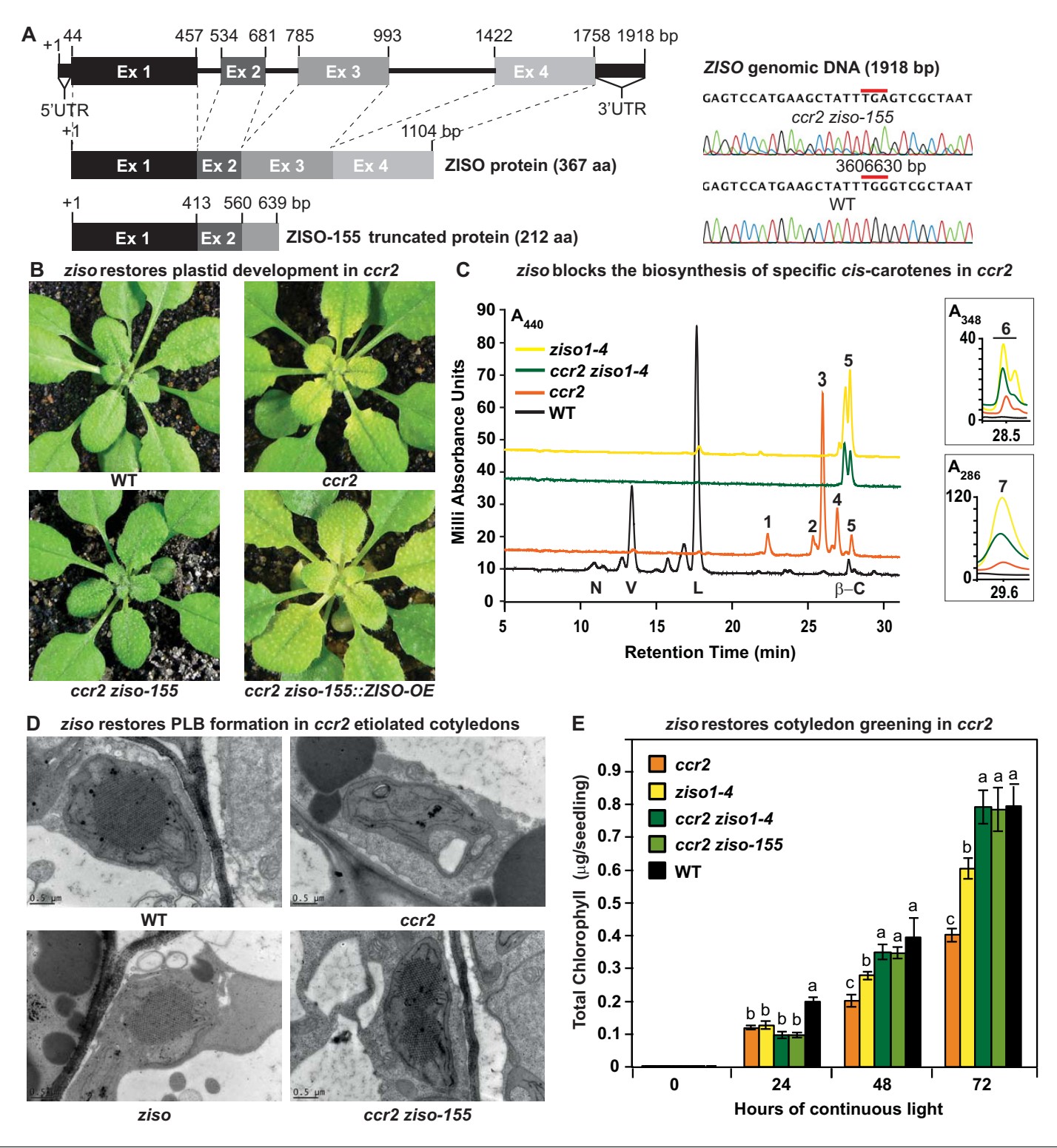

**Figure 4.** *ziso* alters *cis*-carotene profile to restore PLB formation, plastid development and cotyledon greening in *ccr2*. (**A**) Schematic structure of the wild type *ZISO* gDNA, ZISO protein and the truncated version of the *ZISO-155* genomic sequence. *ccr2 ziso-155* contains a G->A mutation in AT1G10830 (3606630 bp) as confirmed by Sanger sequencing that results in a premature stop codon (TGA) in exon 3. (**B**) Rosette images of WT, *ccr2*, *ccr2 ziso-155*, and *ccr2 ziso-155::ZISO-OE#5* showing leaf pigmentations in newly emerged leaves following a reduction in photoperiod. Images are representative of 84/89 $T_4$ generation *ccr2 ziso-155* plants and six independent lines of *ccr2 ziso-155::ZISO-OE*. (**C**) Carotenoid profiles of dark grown cotyledons from WT, *ccr2*, *ziso1-4*, and *ccr2 ziso1-4*. Wavelengths close to the absorption maxima of $A_{440nm}$ (major carotenoids and ζ-carotene isomers),

*Figure 4 continued*

$A_{348nm}$ (phytofluene) and $A_{286nm}$ (phytoene) are shown. Neoxanthin (N); violaxanthin (V); lutein (L); β-carotene (β-C); neurosporene (1 and 2); tetra-*cis*-lycopene (3); pro-neurosporene (4); ζ-carotene (5); phytofluene (6); phytoene (7). (D) Transmission electron micrographs of a representative etioplast from 5-d-old dark grown cotyledons. The etioplasts of WT, *ziso* and *ccr2 ziso-155* show well-developed PLBs, while *ccr2* does not have any. Images are representative of 15 plastids from at least 5 TEM sections. (E) Total chlorophyll levels in cotyledons following de-etiolation. WT, *ccr2*, *ziso1-4*, *ccr2 ziso-155*, and *ccr2 ziso1-4* were grown in darkness for 4 d, exposed to continuous white light and chlorophyll measured at 0, 24, 48 and 72 hr. Letters within a time point denote statistical analysis by ANOVA with a post-hoc Tukey test (n > 20 seedlings). Error bars denote standard error of means (SEM).

responsive to a stimulus (134/382; FDR < $3.7xE^{-7}$) involving hormones and abiotic stress (*Supplementary file 3*). Juvenile leaves of both *ccr2* and *ccr2 ziso-155* showed a significant enrichment in DE genes also responsive to a stimulus (470/1212; FDR < $2.4xE^{-34}$ and 1724/5510; FDR < $5.4xE^{-43}$, respectively) involving several hormones and stress. Even more intriguing was the enhanced enrichment of DE genes specific to *ccr2 ziso-155* juvenile leaves that were involved in biological regulation (1623/5510; FDR < $4.2xE^{-30}$) and epigenetic processes (184/5510; FDR < $3.1xE^{-11}$) such as DNA methylation, histone modification and gene silencing (*Supplementary file 4*).

We utilised Genevestigator to compare DE genes in etiolated seedlings of *ccr2* and *ccr2 ziso-155* with that of mutant germplasm growing on MS media + /- chemical treatments in an attempt to identify co- or contra-regulated changes of gene expression (>20% overlap) (*Supplementary file 3*). Norflurazon, a carotenoid inhibitor of PDS activity and inducer of a retrograde signal(s) was able to induce 30–35% of DE genes in *ccr2*, which was not apparent in *ccr2 ziso-155* (12–14%). This finding was further corroborated by comparison with another published data set (*Page et al., 2017*), where etiolated seedlings of *ccr2* and NF treated de-etiolated seedlings shared a 15–21% overlap in DE genes (*Supplementary file 3*). In contrast, there was no clear overlap in DE genes associated with a brief period of white light following de-etiolation under far red light that induced generation of singlet oxygen, a retrograde signal that obviously regulates a different set of nuclear encoded genes to that of NF (*Page et al., 2017*). An unexpected finding was the DE genes in *ccr2* shared 31–42% in common with the *cop9* and *cop1* mutants. *ccr2 ziso-155* contra-regulated the DE genes in *cop9*, but not those in *cop1*. Genes regulated during light-mediated germination were contra-expressed in *ccr2* (28–48%), yet co-expressed in *ccr2 ziso-155* (44–48%).

We next searched for differentially expressed genes in *ccr2* that were attenuated or contra-expressed in the *ccr2 ziso-155*. Twenty contra-expressed genes were identified to be enriched in process related to photosynthesis, pigment biosynthesis and light stimulus response (5/20; FDR < $1.2xE^{-4}$) (*Supplementary file 5*). However, none of the 20 contra-regulated genes were miss-regulated by a brief period of white light following de-etiolation under far red light, that was shown to be associated with the generation of a singlet oxygen related retrograde signal (*Page et al., 2017*). The expression of *DET1* and *COP1* genes, encoding skotomorphogenesis-associated proteins, were up-regulated in *ccr2*, yet down-regulated in *ccr2 ziso-155* (*Table 2*). This finding is consistent with the fact that DE genes miss-expressed in *ccr2 ziso-155* leaf tissues were enriched in chromatin modifying processes. *det1.1* mutants were shown to have reduced *PIF3* transcripts, and higher HY5 protein levels that activate downstream *PhANG* expression (*Table 2*) (*Lau and Deng,*

**Table 1.** A *cis*-carotene derived ACS acts in parallel to DET1 to control PLB formation.

| Germplasm | Hypocotyl Length (mm) | | Apical hook | Cotyledon | % PLB (-D15) | % PLB (+D15) | *cis*-carotenes |
|---|---|---|---|---|---|---|---|
| WT | Normal | 13.4 ± 0.2 | Yes | Closed | 100 | 100 | None detected |
| *ccr2* | normal | 13.8 ± 0.2 | yes | closed | 0 | 85 | phyt, pflu, ζ-C, p-N, p-Lyc |
| *ccr2 det1-154* | shorter | *8.3 ± 0.2 | no | open | 69 | 0 | reduced *cis*-carotenes |
| *det1-154* | shorter | *9.9 ± 0.1 | no | open | ND | ND | phyt, pflu and ζ-C |

ND; not determined; p-N; pro-neurosporene, p-Lyc; pro-lycopene (tetra-*cis*-lycopene), phyt; phytoene, pflu; phytoflurene, ζ-c; ζ-carotene, *; denotes statistical significance (ANOVA, p<0.05).

**Table 2.** Contra-regulated differential gene expression in etiolated seedlings and young leaves of *ccr2 ziso-155.*

| Gene id | GENE | PhANG | Protein encoding description | Etiolated seedlings | | Young leaves | | det1-1 | NF-1 | NF-2 |
|---|---|---|---|---|---|---|---|---|---|---|
| | | | | ccr2 | ccr2 ziso-155 | ccr2 | ccr2 ziso-155 | | | |
| At1g09530 | *PIF3* | | Transcription factor interacts with photoreceptors and negatively regulates signalling | 30 | 0.1 | 220 | 0.1 | ↓ | −5.0 | NS |
| At4g10180 | *DET1/FUS2* | | Encodes a nuclear-localized protein repressor of photomorphogenesis | 5.1 | 0.1 | 5.9 | 0.2 | NS | NS | NS |
| At3g19390 | | | Granulin repeat cysteine protease family protein | 4.4 | NS | 6.8 | NS | NS | NS | NS |
| At5g13210 | | | Unknown conserved expressed protein | 3.8 | NS | 0.4 | NS | ↑ | NS | NS |
| At3g45730 | | | Unknown expressed protein | 2.8 | NS | 2.4 | NS | NS | NS | 10.6 |
| At5g43500 | *ATARP9* | | Encodes an expressed protein similar to actin-related proteins | 2.4 | NS | 2.2 | NS | NS | NS | NS |
| At5g48240 | | | Unknown expressed protein | 2.1 | NS | 2.2 | NS | NS | NS | NS |
| At2g32950 | *COP1/FUS3* | | Repressor of photomorphogenesis and induces skotomorphogenesis | 2.0 | 0.0 | 8.9 | 0.1 | ↑ | NS | NS |
| At5g11260 | *HY5* | | Transcription factor negatively regulated by COP1, promotes light responsive gene expression | 0.5 | 8.1 | 0.3 | 8.4 | NS | NS | 2.8 |
| At4g02770 | *PSAD1* | | Expressed protein with similarity to photosystem I subunit II | 0.5 | NS | 0.5 | NS | ↑ | −12.3 | 0.15 |
| At3g17070 | | | Peroxidase family expressed protein | 0.5 | NS | 0.5 | NS | NS | NS | NS |
| At2g31751 | | | Potential natural antisense gene, expressed protein | 0.4 | NS | 0.5 | NS | NS | NS | NS |
| At4g15560 | *DXS/CLA1* | yes | 1-deoxyxylulose 5-phosphate synthase activity in MEP pathway | 0.3 | 4.2 | 0.1 | 16.2 | NS | NS | 0.42 |
| At4g34350 | *ISPH/CLB6* | yes | 4-hydroxy-3-methylbut-2-enyl diphosphate reductase in MEP pathway | 0.3 | 9.4 | 0.2 | 11 | ↑ | NS | NS |
| At1g24510 | *TCP-1* | | T-complex expressed protein one epsilon subunit | 0.3 | 12.0 | 0.1 | 7.9 | NS | NS | NS |
| At3g59010 | *PME35* | | Pectin methylesterase that regulates the cell wall mechanical strength | 0.2 | NS | 0.4 | NS | ↓ | NS | NS |
| At1g29930 | *CAB1/LHCB1.3* | yes | Subunit of light-harvesting complex II (LHCII), which absorbs light | 0.2 | 13 | 0.2 | 11 | NS | NS | NS |
| At2g05070 | *LHCB2.2* | yes | Light-harvesting chlorophyll a/b-binding (LHC) protein that constitute the antenna system | 0.2 | NS | 0.2 | NS | ↑ | −3.6 | NS |
| At5g13630 | *GUN5/CHLH* | yes | Magnesium chelatase involved in plastid-to-nucleus signalling | 0.2 | 17 | 0.2 | 20 | ↑ | −3.3 | 0.33 |
| At1g67090 | *RBCS1a* | yes | Member of the Rubisco small subunit (RBCS) multigene family functions in photosynthesis | 0.1 | 67 | 0.1 | 61 | NS | NS | NS |

Notes: NS; not significant. Transcriptomic data; *det1-1* (**Schroeder et al., 2002**), norflurazon (NF-1; **Page et al., 2017**), norflurazon (NF-2; **Koussevitzky et al., 2007**), *PhANG*; *Photosynthesis associated nuclear gene.* Numbers refer to fold change relative to WT = 0 (except for NF-1 where positive and negative numbers indicate up and down-regulation, respectively relative to WT = 1.

*2012*). A lower PIF3/HY5 ratio can be associated with *PhANG* expression. Indeed, our comparative analysis of contra-expressed genes in *ccr2 ziso-155* revealed the down-regulation of *PIF3*, up-regulation of *HY5* and *PhANG* expression (e.g. *LHCB2; LIGHT-HARVESTING CHLOROPHYLL B-BINDING 2*) (**Table 2**). This would lead to a lower PIF3/HY5 ratio in *ccr2 ziso-155* when compared to WT. It is

not unusual to observe miss-regulation of *PhANG* expression in mutants having impaired plastid development (*Ruckle et al., 2007*; *Woodson et al., 2011*). However, when we compared the 20 contra-regulated gene list with two different published data sets that reported miss-regulation of genes following NF treatment of de-etiolated seedlings (*Koussevitzky et al., 2007*; *Page et al., 2017*), only three *PhANGs* (*LHCB2.2*, *PSI-D* and *DXS*) and *GUN5* were down-regulated in *ccr2* (*Supplementary file 5*). Further comparison of the 20 contra-regulated genes with DE genes miss-regulated in the *gun5* mutant revealed no similarity. This highlights that *ccr2* regulates a unique set of genes not related to a NF generated or GUN5 mediated retrograde signal. In summary, the repression of negative regulators of photomorphogenesis, correlates well with the up-regulation of *PhANG* expression in *ccr2 ziso-155* and links *cis*-carotene accumulation to the regulation of a unique gene set involved in mediating plastid development.

## Activation of photomorphogenesis by *det1-154* restores plastid development in *ccr2*

We searched the SNP deserts of the remaining twenty-four *rccr2* lines for genes that could link *cis*-carotene signalling to regulators of photomorphogenesis. $rccr2^{-154}$ was mapped to a causal mutation in de-*etiolated 1* (*det1*), hereafter referred as *ccr2 det1-154*, which restored plastid development in immature *ccr2* leaves (*Figure 3*). Sequencing of the *det1-154* genomic DNA identified a G to A point mutation at the end of exon 4. Sequencing of the *det1-154* cDNA revealed the removal a 23 amino acid open reading frame due to alternate splicing (*Figure 5A*). Quantitative PCR analysis confirmed that the shorter *DET1-154* transcript (spliced and missing exon 4) was highly enriched (approx. 200 fold) in *ccr2 det1-154*, while the normal *DET1* transcript (which contains exon 4) was repressed in *ccr2 det1-154* (*Figure 5—figure supplement 1A*). The phenotypes of *ccr2 det1-154* and *det1-154* were intermediate to that of *det1-1* (*Chory et al., 1989*), showing a smaller rosette with a shorter floral stem height and reduced fertility relative to the WT (*Figure 5—figure supplement 1B*). The overexpression of the full length *DET1* transcript (*CaMV35s::DET1-OE)* in *ccr2 det1-154* restored the virescent phenotype in *ccr2* leaves from plants grown under an 8 hr photoperiod (*Figure 5B*). Therefore, alternative splicing of *det1* and removal of exon four appeared sufficient to restore plastid development in *ccr2* leaves grown under a shorter photoperiod.

We investigated how *det1-154* can restore plastid development in *ccr2*. *ccr2 det1-154* mature leaves contained less carotenoids and chlorophylls compared to *ccr2* (*Figure 5—figure supplement 1C*). That is, the xanthophylls and β-carotene were all significantly reduced by *det1-154*. *det1-154* also reduced total *cis*-carotene content in *ccr2* etiolated cotyledons (*Figure 5C*; *Figure 5—figure supplement 1D*). That is, di-*cis*-ζ-carotene, pro-neurosporene and tetra-*cis*-lycopene were significantly reduced in *ccr2 det1-154*, while phytoene, phytofluene and tri-*cis*-ζ-carotene levels were not significantly different to *ccr2* (*Figure 5—figure supplement 1D*). *ccr2* prevented PLB formation during skotomorphogenesis, yet displayed an apical hook with closed cotyledons and normal hypocotyl length, that did not resemble *det1* photomorphogenic mutants (*Table 1*). TEM confirmed that the dark-grown cotyledons from etiolated *ccr2 det1-154* seedlings showed PLBs in 69% of etioplasts examined during skotomorphogenesis (*Table 1*; *Figure 5—figure supplement 1E*). The restoration of a PLB in *ccr2 det1-154* dark grown seedlings coincided with a restoration of cotyledon greening following de-etiolation (*Figure 5E*). In leaves and etiolated cotyledons, *det1* mutants exhibited reduced total carotenoid and/or chlorophyll content when compared to WT (*Supplementary file 6*). That is, the xanthophylls and β-carotene were all significantly reduced in *det1* mutants. We detected traces of phytoene and phytofluene in emerging leaves and in addition tri-*cis*-ζ-carotene at higher levels in etiolated cotyledons of *det1* mutants (*Supplementary file 6*). *det1-154* activated photomorphogenesis in *ccr2* as evident by etiolated seedlings having characteristic shorter hypocotyl, no apical hook and opened large cotyledons similar to *det1-1* (*Figure 5D*). Therefore, the reduction of the full length *DET1* mRNA in *ccr2* caused a reduction in specific *cis*-carotenes (di-*cis*-ζ-carotene, pro-neurosporene and tetra-*cis*-lycopene) and restored PLB formation (*Table 1*).

## D15 inhibition of carotenoid cleavage activity reveals a *cis*-carotene cleavage product that controls PLB formation

Can the accumulation of specific *cis*-carotenes directly perturb PLB formation as hypothesised (*Park et al., 2002*), or does production of an apocarotenoid signal regulate PLB formation? We

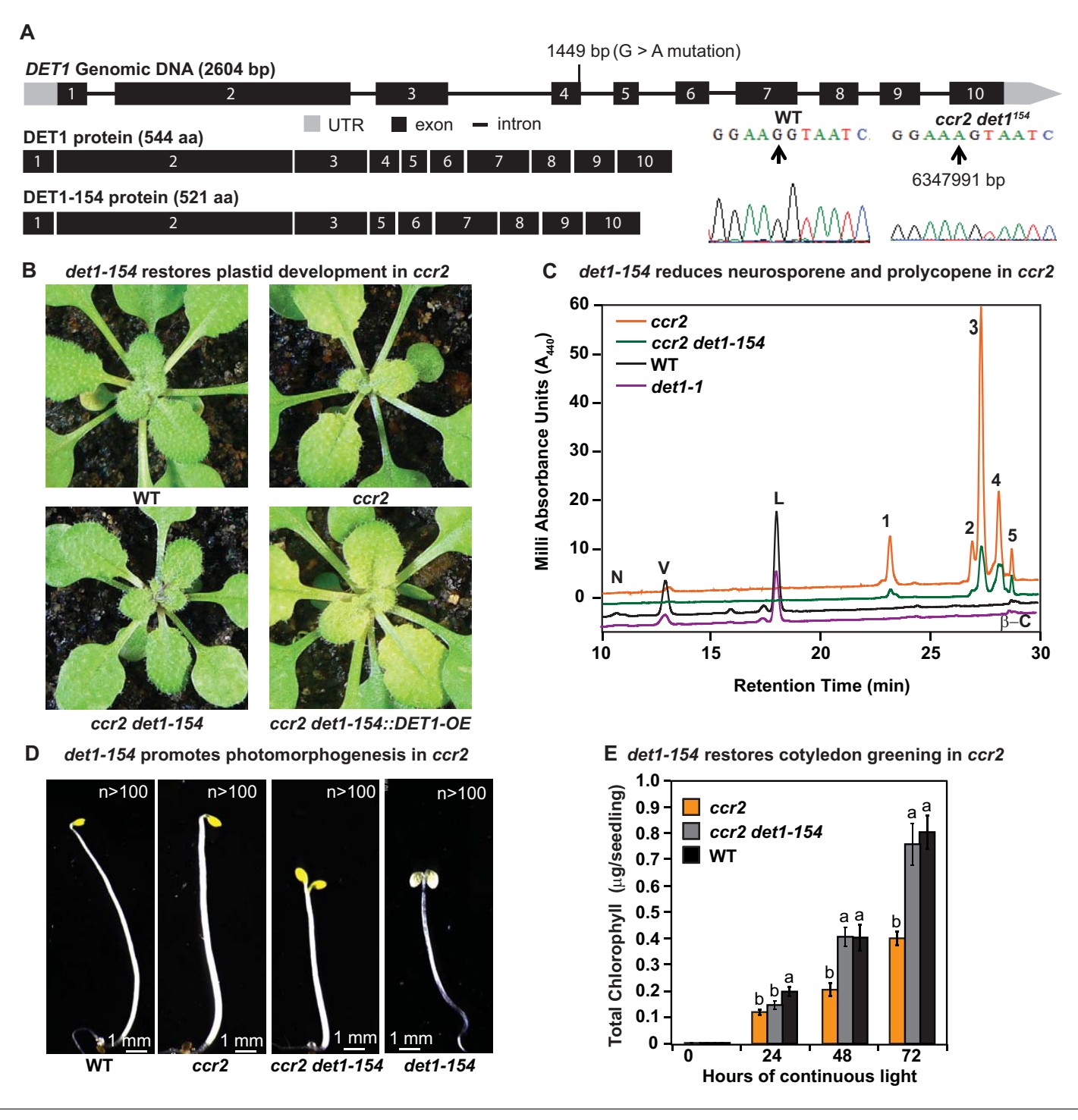

**Figure 5.** *det1* restores PLB formation, plastid development and cotyledon greening in *ccr2*. (**A**) Schematic structure of the wild type *DET1* gDNA, DET1 protein and alternative spliced *DET1-154* protein. A G->A mutation at the end of exon 4 (1449 bp) of AT4G10180 (6347991 bp) was confirmed by Sanger sequencing that leads to the skipping of exon 4 (69 bp). The *DET1-154* splice variant produces a shorter protein (521 aa). Exon 4 comprises 23 amino acids in-frame, having homology to the six-hairpin glycosidase-like (IPR008928) domain. (**B**) Rosette images of WT, *ccr2*, *ccr2 det1-154*, and *ccr2 det1-154::DET1-OE* showing leaf pigmentations in newly emerged leaves from plants shifted from a 16 hr photoperiod (2 weeks old) to an 8 hr photoperiod for 1 week. Images are representative of 122/149 T$_1$ generation *ccr2 det1-154* plants from 12 independent lines surviving Basta herbicide selection after being transformed with pEARLEY::*DET1-OE*. (**C**) Carotenoid profiles of 7-d-old dark grown cotyledons from WT, *ccr2*, *ccr2 det1-154* and *det1-1* etiolated seedlings. Wavelengths close to the absorption maxima of A$_{440}$ (major carotenoids and ζ-carotene isomers) show neoxanthin (N); violaxanthin (V); lutein (L), β-carotene (β-C) in WT and neurosporene isomers (1 and 2) tetra-*cis*-lycopene (3); pro-neurosporene (4), and pro-ζ-carotene

*Figure 5 continued on next page*

Figure 5 continued

(5) in *ccr2* and to a less extent in *ccr2 det1-154*. (D) Etiolated seedling morphology of WT, *ccr2*, *ccr2 det1-154* and *det1-154*. Seedlings were grown in the dark for 7 d on MS media without sucrose. Representative images (>100 seedlings from independent experiments) depict a typical apical hook for WT and *ccr2*, and shorter hypocotyl with open cotyledons for *ccr2 det1-154* and *det1-154*. (E) Chlorophyll levels in cotyledons following de-etiolation. *ccr2*, *ccr2 det1-154* and WT were etiolated for 4 d in darkness and thereafter exposed to continuous white light. Chlorophyll measurements were taken at 0, 24, 48 and 72 hr after de-etiolation. Letters within a time point denote statistical analysis by one-way ANOVA with a post-hoc Tukey test (n > 20 seedlings). Error bars denote standard error of means.

The online version of this article includes the following figure supplement(s) for figure 5:

**Figure supplement 1.** *det1-154* has alternative splicing and reduced pigments, *cis*-carotenes and restored PLB formation in *ccr2*.

crossed *ccr2* to carotenoid cleavage dioxygenase loss-of-function mutants; *ccd1*, *ccd4*, *ccd7* (*max3*) and *ccd8* (*max4*) and tested if plants exposed to a shorter photoperiod would revert the virescent leaf phenotype of *ccr2*. We analysed more than 10 plants for each of the *ccr2 ccd* double mutant lines and observed a perturbation in plastid development in >93% of plants, each displaying clearly visible yellow virescent leaves similar to *ccr2* (*Figure 6—figure supplement 1A–B*). We concluded that no single *ccd* mutant was sufficient to block the production of any *cis*-carotene derived cleavage product. However, there is a degree of functional redundancy among family members, as well as multiple cleavage activities and substrate promiscuity (*Hou et al., 2016*).

To address the challenge of CCD functional redundancy and substrate promiscuity we decided to utilise the aryl-C3N hydroxamic acid compound (D15), which is a specific inhibitor (>70% inhibition) of 9,10 cleavage enzymes (CCD) rather than 11,12 cleavage enzymes (NCED) (*Figure 1—figure supplement 1A*) (*Sergeant et al., 2009*; *Van Norman et al., 2014*). We imaged etioplasts from WT and *ccr2* etiolated seedlings treated with D15 (*Van Norman et al., 2014*). The majority (86%) of D15-treated *ccr2* etioplasts displayed a PLB, whilst in control treatments *ccr2* etioplasts showed no discernible PLB (*Figure 6A*; *Supplementary file 2*). Total PChlide levels in WT and *ccr2* before and after D15 treatment were similar (*Figure 6B*). As expected, etiolated *ccr2* seedlings grown on D15-treated MS media accumulated chlorophyll in cotyledons within 24 hr of continuous light treatment following de-etiolation in a manner similar to WT (*Figure 6C*). D15 significantly enhanced di-*cis*-ζ-carotene and pro-neurosporene, yet reduced tetra-*cis*-lycopene in etiolated cotyledons of *ccr2* (*Figure 6D*). In WT etiolated cotyledons, D15 significantly enhanced violaxanthin, neoxanthin and antheraxanthin content, which was previously shown to occur in Arabidopsis roots (*Van Norman et al., 2014*) (*Figure 6E*). Treatment of dark and light grown wild type seedlings with D15 did not cause adverse pleiotropic effects on cotyledon greening (*Figure 6*), hypocotyl elongation (data not shown) or plastid development in cotyledons (*Table 1*, *Supplementary file 2*). Therefore, apocarotenoid formation from either cleavage of di-*cis*-ζ-carotene and/or pro-neurosporene in *ccr2* can perturb PLB formation independent of PChlide biosynthesis.

## A *cis*-carotene cleavage product promotes *POR* transcription in *det1-154*

We searched for the regulatory mechanism by which a *cis*-carotene cleavage product could control POR regulation and hence PLB formation during skotomorphogenesis. *PORA* transcript levels are relatively high in etiolated seedlings, becoming down-regulated upon exposure to white light or when photomorphogenesis is activated (*Armstrong et al., 1995*; *Sperling et al., 1998*). Reduced *PORA* expression will perturb PLB formation, while a lack of *PORA* expression in *det1-1 or cop1* mutants will block PLB formation, and overexpression of *PORA* can restore PLB formation (*Sperling et al., 1998*; *Paddock et al., 2012*). *PORA* transcript levels were similar in WT and *ccr2* etiolated tissues, even though *ccr2* lacked a PLB (*Figure 7A*). There was a substantial reduction in *PORA* mRNA expression in *det1-154*, as previously shown for other *det1* mutant alleles that lack a PLB. Interestingly, *PORA* transcript levels were restored to WT levels in *ccr2 det1-154* (*Figure 7A*, *Figure 5—figure supplement 1E*). D15 treatment did not affect *PORA* transcript levels in WT, *ccr2* or *det1-154*, however significantly repressed *PORA* expression in *ccr2 det1-154* back to *det1-154* levels (*Figure 7A*). Therefore, the *ccr2* generated *cis*-carotene cleavage product can override the negative regulation of *PORA* transcription enabled by *det1-154*, yet does not alter *PORA* expression when compared to WT (*Figure 7E*).

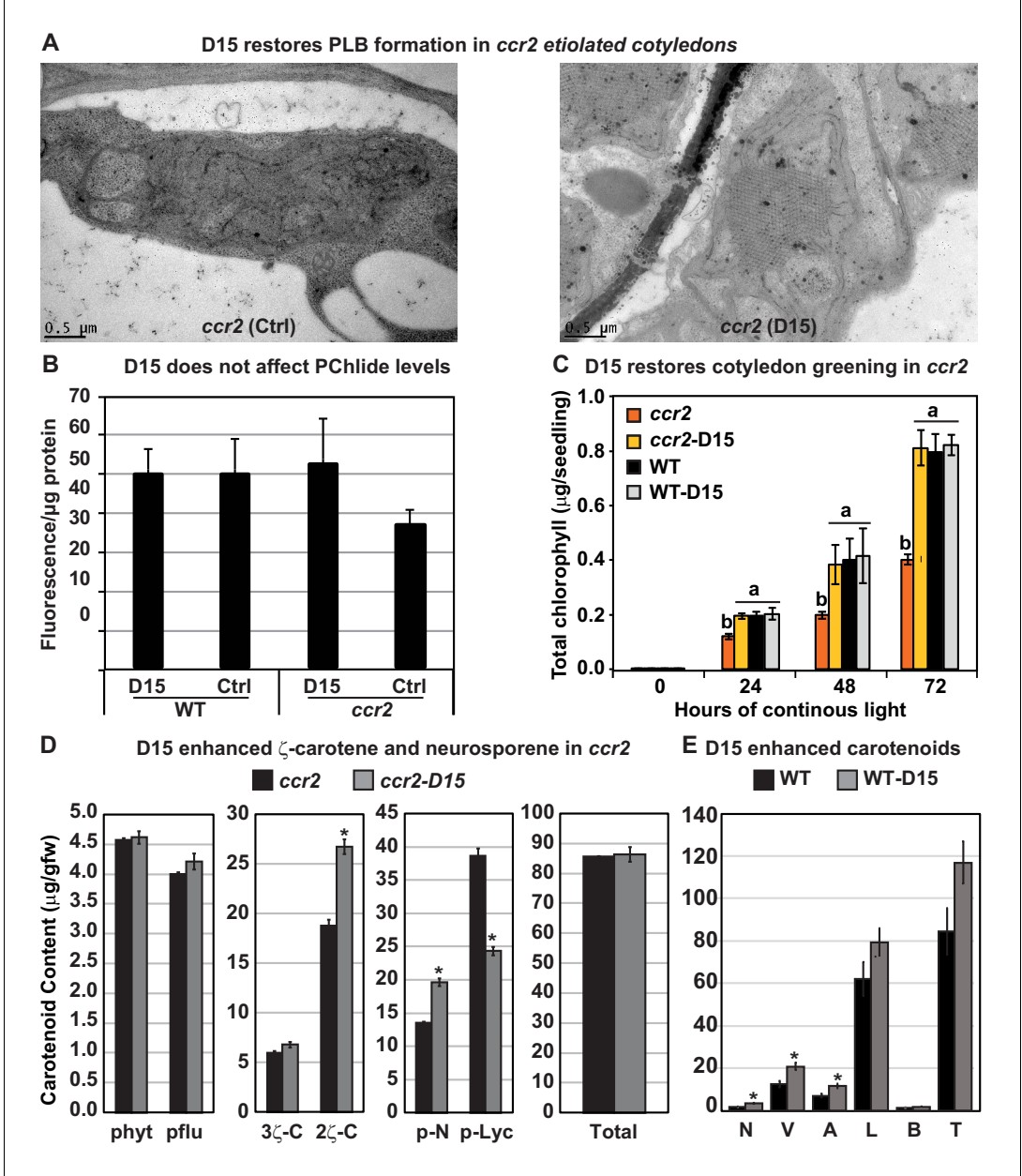

**Figure 6.** The carotenoid cleavage dioxygenase (CCD) inhibitor, D15, restores PLB formation in etiolated *ccr2* seedlings, cotyledon greening following de-etiolation and alters *cis*-carotene accumulation. (**A**) Transmission electron micrographs of a representative etioplast from 5-d-old dark grown cotyledons reveal a well-developed PLB in *ccr2* treated with the D15, but not in *ccr2* treated with ethanol only (control; ctrl). (**B**) Pchlide levels in Wild Type (WT) and *ccr2* treated + / - D15. Fluorescence was measured at 638 nm and 675 nm with an excitation at 440 nm. Net fluorescence of Pchlide was calculated and normalised to protein content. (**C**) D15 restores chlorophyll accumulation in *ccr2* de-etiolated seedlings exposed to continuous light. Twenty seedlings from each of three biological replicates were harvested for chlorophyll determination in every 24 hr under continuous light. Statistical analysis was by ANOVA with a post-hoc Tukey test (n = 20 seedlings). (**D**) *cis*-carotene quantification in etiolated cotyledons of *ccr2* treated with D15. phytoene (phyt), phytofluene (pflu), tri-*cis*-ζ-carotene (3ζ-C), di-*cis*-ζ-carotene (2ζ-C), pro-neurosporene (p-N), tetra-*cis*-lycopene (p-lyc) and total *cis*-carotenes were quantified at absorption wavelengths providing maximum detection. Star denotes significance (ANOVA, p<0.05). Error bars show standard error (n = 4). (**E**) Quantification of carotenoid levels in etiolated tissues of WT treated with D15. Neoxanthin (N); violaxanthin (V); antheraxanthin (A), lutein (L), β-carotene (β-C) and total carotenoids (T) were quantified at a 440 nm absorption wavelength providing maximum detection. Star denotes significance (ANOVA, p<0.05). Data is representative of two independent experiments.

The online version of this article includes the following figure supplement(s) for figure 6:

**Figure supplement 1.** The loss-of-function in individual members of the *carotenoid cleavage dioxygenase* gene family cannot restore plastid development in *ccr2* rosettes.

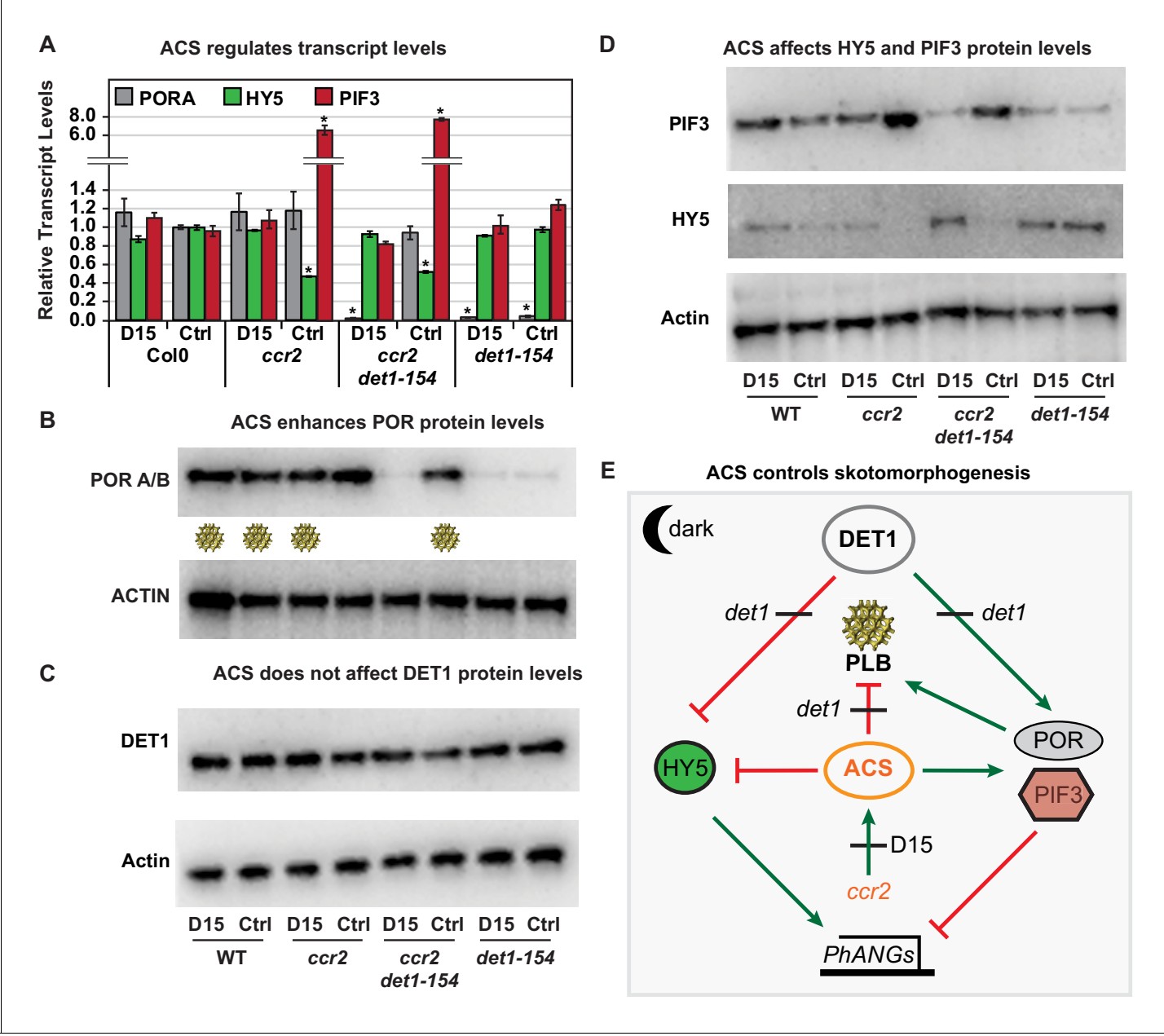

**Figure 7.** Chemical inhibition of CCD activity revealed how a *ccr2* generated apocarotenoid signal transcriptionally up-regulates POR and PIF3 in parallel to *det1-154* during skotomorphogenesis. (A) Transcript levels of *PORA, PIF3* and *HY5* in WT, *ccr2, ccr2 det1-154* and *det1-154* etiolated seedlings growing on MS media (+ /- D15). Statistical analysis denoted as a star was performed by a pair-wise *t*-test (p<0.05). Error bars represent standard error of means. (B), (C) and (D) Representative western blot images showing POR, DET1, PIF3 and HY5 protein levels, respectively. Proteins were extracted from WT, *ccr2* and *ccr2 det1-154* etiolated seedlings grown on MS media without (control; Ctrl) or with the chemical inhibitor of CCD activity (D15). The membrane was re-probed using anti-Actin antibody as an internal loading control. Lattice-like symbol below POR western (B), represents formation of a PLB in etiolated cotyledons from that genotype and treatment. (E) Model describing how a *cis*-carotene derived cleavage product, ACS, regulates POR, HY5, PIF3 and PLB formation during skotomorphogenesis. DET1 maintains skotomorphogenesis by post-transcriptionally maintaining a higher and lower PIF3 and HY5 protein levels, respectively. HY5 promotes and PIF3 represses *PhANG* expression. *det1* mutants trigger photomorphogenesis in that they lack *POR* mRNA transcripts, protein and a PLB. *ccr2* generates ACS that enhances *POR* mRNA transcript and protein levels that enable PLB formation in *det1-154*. *det1-154* restores PLB formation in *ccr2* by blocking a signalling pathway acting independent of POR. The online version of this article includes the following figure supplement(s) for figure 7:

**Figure supplement 1.** The DET1-154 peptide is smaller in *det1-154* mutant genotypes.

We analysed the POR protein levels in dark grown seedlings, with or without D15, noting that wild-type and *ccr2* accumulate POR (*Park et al., 2002*), while *det1* lacks POR (*Sperling et al., 1998*) (*Figure 1—figure supplement 1B*). Under the electrophoresis conditions used herein, the Arabidopsis PORA/B proteins were detected as a single immune-reactive signal (PORA; 37 kDa, and PORB; 36 kD) (*Sperling et al., 1998*; *Park et al., 2002*; *Paddock et al., 2012*) (*Figure 7B*). While WT, *ccr2* and *ccr2 det1-154* accumulated wild-type levels of POR protein, *det1-154* accumulated very low and barely detectable levels of POR protein (*Figure 7B*). This was consistent with a lack of POR observed in etiolated tissues from other *det1* alleles (*Sperling et al., 1998*). D15 did not substantially affect POR protein levels in WT, *ccr2* or *det1-154*. However, treatment of *ccr2 det1-154* with D15 reduced POR protein to an undetectable level (*Figure 7B and E*) and this was not due to *ccr2* or D15 significantly changing DET1 protein levels (*Figure 7C*). Extended gel electrophoresis revealed that the size of the DET1-154 peptide (59 kDa) was indeed smaller in comparison to DET1 (64 kDa) due to the splicing of exon 4 (*Figure 7—figure supplement 1*). Therefore, *cis*-carotene cleavage in *ccr2* generated a signal that can block *det1-154* mediated repression of *PORA* transcription and restore WT POR protein levels and PLB formation in *det1-154*.

## A *cis*-carotene cleavage product acts independent of DET1 to regulate PIF3 and HY5 during skotomorphogenesis

DET1 is a negative regulator of photomorphogenesis, such that *det1* mutants lack PIF3 protein and accumulate higher HY5 protein levels during skotomorphogenesis according to published results (*Osterlund et al., 2000*; *Dong et al., 2014*) (see *Figure 1—figure supplement 1B*). The miss-regulation or loss-of-function in *PIF3* or *HY5* does not block PLB formation and skotomorphogenesis per se (*Chang et al., 2008*; *Stephenson et al., 2009*; *Liu et al., 2017*). We investigated if the apocarotenoid signal can affect the PIF3-HY5 regulatory hub during skotomorphogenesis. The transcript levels of *PIF3* and *HY5* in *ccr2* and *ccr2 det1-154* etiolated tissues was substantially higher (>6 fold) and lower (>50%), respectively (*Figure 7A*). The same trend was observed in our transcriptomic analysis of *ccr2* etiolated tissues (*Table 2*). D15 treatment restored *HY5* and *PIF3* mRNA expression back to WT levels in *ccr2* and *ccr2 det1-154* (*Figure 7A*). The expression of these two genes was not significantly different in *det1-154* compared to WT, regardless of D15 treatment. Therefore, a *ccr2* generated *cis*-carotene cleavage product can transcriptionally regulate *HY5* and *PIF3* (*Figure 7E*).

We next examined the levels of PIF3 and HY5 protein during skotomorphogenesis. It should be noted that wild-type had higher levels of PIF3 and very low or trace levels of HY5 in etiolated tissues, with the converse in *det1-154* (*Figure 7D and E*), a result consistent with previous reports (*Figure 1—figure supplement 1B*). In contrast, *PIF3* and *HY5* transcript levels were similar in WT and *det1-154* revealing that *det1-154* post-transcriptionally regulated PIF3 and HY5 protein levels (*Figure 7A and D*). In both *ccr2* and *ccr2 det1-154* etiolated cotyledons, PIF3 protein levels were considerably higher, while HY5 protein levels were undetected, a trend consistent with the relative change in transcript levels (*Figure 7A and D*). D15 treatment reverted PIF3 and HY5 protein levels in *ccr2* and *ccr2 det1-154* back to WT and *det1-154* levels, respectively. D15 did not affect PIF3, HY5 or DET1 protein levels in WT. This indicates that an apocarotenoid signal can transcriptionally alter the PIF3/HY5 ratio in the presence or absence of DET1, indicating it acted independent and either in parallel with, or downstream of DET1. The relative difference in PIF3 protein levels in *ccr2* compared to *ccr2 det1-154* in the presence of D15 would suggest the two pathways operate in parallel.

## *cis*-carotene cleavage in *ccr2* regulates PhANG expression during photomorphogenesis

PIF3 and HY5 are key regulatory transcription factors involved in controlling the dark to light transition (*Osterlund et al., 2000*; *Dong et al., 2014*). PIF3 and HY5 protein levels decrease and increase, respectively thereby activating *PhANG* expression that facilitates differentiation of an etioplast into a chloroplast. We investigated if a *ccr2* generated apocarotenoid signal regulated the PIF3/HY5 regulatory hub and *PhANG* expression during photomorphogenesis. The transcript levels of *PIF3* and *HY5* in *ccr2* and *ccr2 det1-154* de-etiolated seedlings (4-d darkness, exposed to 3-d continuous light) were substantially higher (>16 fold) and significantly lower (>40%), respectively (*Figure 8A*). The same trend was observed in our transcriptomic analysis of virescent *ccr2* leaf tissues grown under a short photoperiod (*Table 2*) and dark grown etiolated cotyledons (*Figure 7A*). D15 treatment

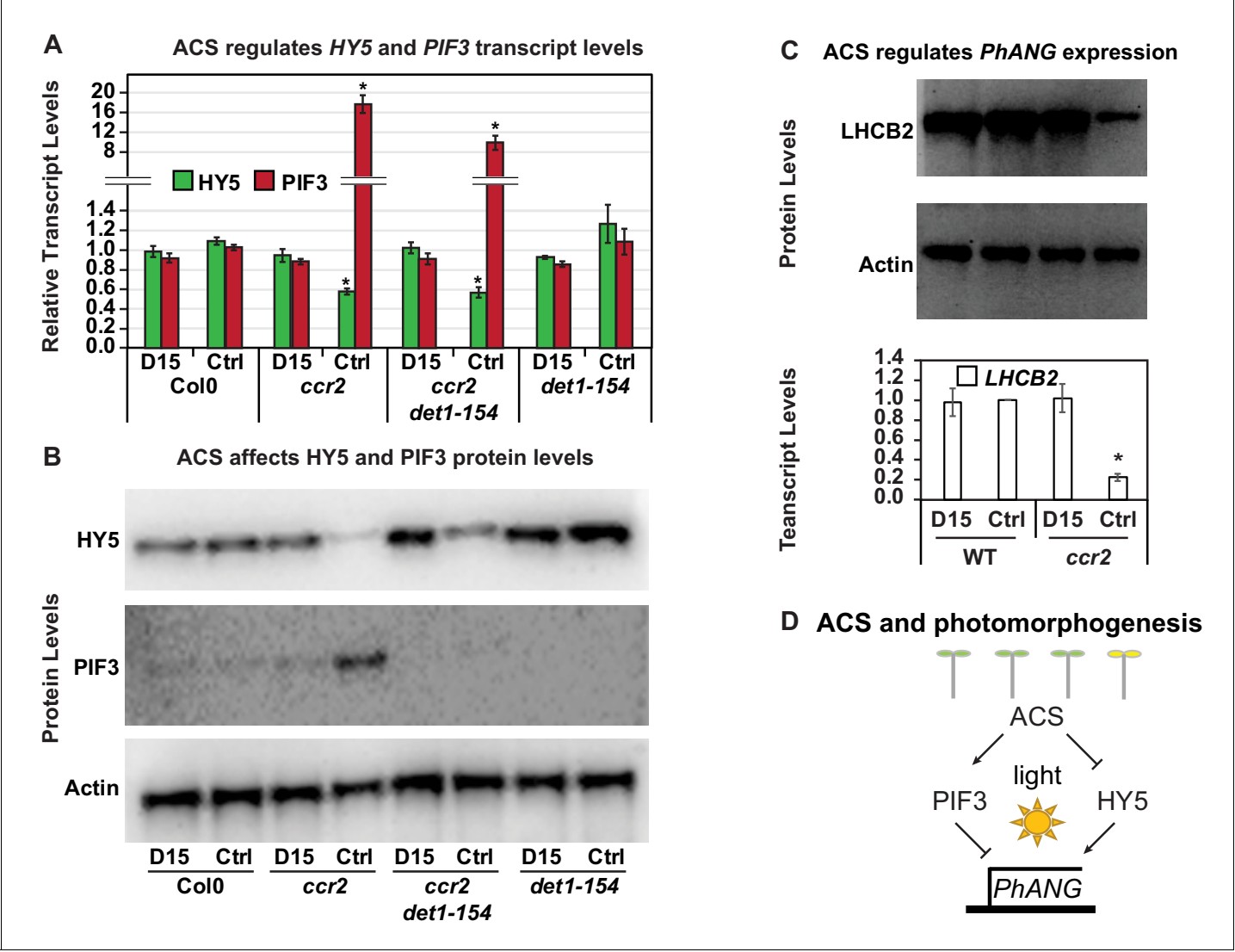

**Figure 8.** Chemical inhibition of CCD activity revealed how a *ccr2* generated apocarotenoid signal transcriptionally represses HY5 and *LHCB2* expression during photomorphogenesis. (**A**) Transcript levels of *PIF3* and *HY5* in WT, *ccr2*, *ccr2 det1-154* and *det1-154* de-etiolated seedlings growing on MS media + /- D15. (**B**) Representative western blot images showing PIF3 and HY5 protein levels in WT, *ccr2*, *ccr2 det1-154* and *det1-154* de-etiolated seedlings growing on MS media + /- D15. The membrane was re-probed using anti-Actin antibody as an internal loading control. (**C**) Protein and transcript levels of *LHCB2* expression in WT and *ccr2* de-etiolated seedlings growing on MS media + /- D15. (**D**) Model showing how ACS regulates HY5 and *LHCB2* expression in *ccr2*. Images of seedlings represent are cotyledons are coloured green or yellow to reflect the delay in chlorophyll biosynthesis induced by ACS as evidenced in *Figure 6c*. De-etiolation of seedlings was performed by transferring 4-d-old etiolated seedlings to continuous light for 3 d to induce photomorphogenesis. Statistical analysis denoted as a star was performed by pair-wise *t*-test (p<0.05). Error bars represent standard error of means. Ctrl; Control; Ctrl, D15; chemical inhibitor of CCD activity.

restored *PIF3* and *HY5* mRNA expression back to WT levels in *ccr2* and *ccr2 det1-154* (*Figure 8A*). The *PIF3* and *HY5* mRNA expression levels were not significantly different in *det1-154* compared to WT, regardless of D15 treatment. The protein levels of PIF3 and HY5 were consistent with their relative gene expression levels in *ccr2*. That is, PIF3 and HY5 protein levels were higher and lower respectively in *ccr2*, and D15 restore their expression back to WT levels (*Figure 8B*). As expected *det1-154* had higher HY5 protein levels compared to WT, and PIF3 was not detectable. D15 had no effect on HY5 or PIF3 protein levels in WT or *det1-154*, however it did enhance HY5 levels in *ccr2 det1-154*. *LHCB2* mRNA and protein expression was significantly reduced in *ccr2*, and was restored back to WT expression levels by D15 treatment (*Figure 8C*). The reduction in *LHCB2* gene expression was consistent with our transcriptomic analysis in virescent leaf tissues of *ccr2* (*Table 2*). In

summary, a *ccr2* generated *cis*-carotene cleavage product can transcriptionally enhance the PIF3/HY5 ratio during photomorphogenesis thereby reducing *PhANG* expression and greening of *ccr2* seedlings (*Figure 8D*).

## Discussion

Plastid and light signalling coordinate leaf development under various photoperiods, and younger leaves display a greater plasticity to modulate their pigment levels in response to environmental change (*Lepistö and Rintamäki, 2012*; *Dhami et al., 2018*). We attribute *ccr2* leaf viresence to the fine-tuning of plastid development in leaf primordia cells as a consequence of *cis*-carotene accumulation and not the generation of singlet oxygen (*Kato et al., 2009*; *Chai et al., 2011*; *Han et al., 2012*; *Page et al., 2017*). Far red light treatment of etiolated seedlings represses PORA activity, while the synthesis of Pchlide continues without conversion into chlorophyllide. Exposure of the pre-treated seedlings to white light generates singlet oxygen and a block in seedling greening (*Page et al., 2017*). Since prolonged dark grown *ccr2* tissues and seedlings exposed to a brief period of white light following de-etiolation under far red light treatment regulate a different set of genes, we deduce that *ccr2* leaf virescence was not due to singlet oxygen generation. Our evidence revealed that leaf virescence was linked to the hyper-accumulation of specific *cis*-carotenes since, *ziso-155* and *det1-154* as well as D15 were able to reduce *cis*-carotene biosynthesis in *ccr2* tissues, andrestore leaf greening in plants grown under a shorter photoperiod (*Figures 4* and *5*). A shorter photoperiod triggered *cis*-carotene hyper-accumulation in newly emerged photosynthetic tissues when CRTISO activity was perturbed and caused leaf virescence (*Figure 9*). The altered plastid development in etiolated cotyledons and younger virescent leaves from *ccr2* cannot be attributed to a block in lutein, strigolactone, ABA or alteration in xanthophyll composition (*Figure 2*). Phytoene, phytofluene and to a lesser extent ζ-carotene were noted to accumulate in wild type tissues from different plant species (*Alagoz et al., 2018*). We also detected traces of these *cis*-carotenes in newly emerged tissues from wild type, and even more so in *det1* mutant leaves. Without the signal itself to assess the physiological function in wild-type plant tissues, we provided evidence for the existence

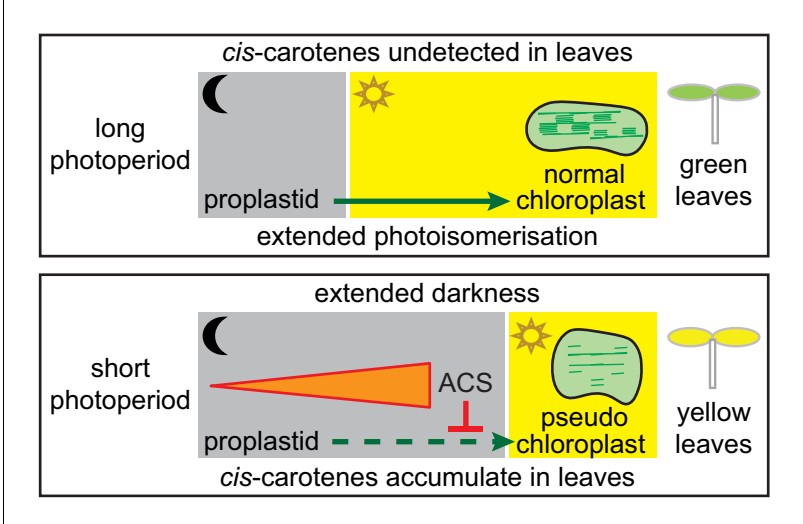

**Figure 9.** Model showing how a *cis*-carotene derived apocarotenoid cleavage product controls plastid development in leaves from plants growing under a shorter photoperiod. Shorter photoperiods that have an extended period of darkness, cause *cis*-carotenes to accumulate in leaf tissues from plants having impaired or lacking carotenoid isomerase activity. Plants growing under a longer photoperiod are exposed to an extended period of photoisomerisation, which stops *cis*-carotene from accumulating to detectable levels. A *cis*-carotene derived apocarotenoid signal (ACS) can perturb proplastid to chloroplast development, leading to the formation of a pseudo-chloroplast with poorly defined thylakoid and grana stacks. As a result, a yellow leaf virescence phenotype becomes visible in newly emerged leaves from carotenoid isomerase mutant plants growing under a shorter photoperiod.

of a *cis*-carotene cleavage product in *ccr2* that can regulate PLB formation during skotomorphogenesis and plastid development during leaf greening independent of, and capable of compensating for mutations in DET1. We contrast how the *cis*-carotene derived novel apocarotenoid signal can transcriptionally control repressor and activator proteins in parallel with DET1, to mediate *PhANG* expression and plastid development (**Figure 7E**).

## A *cis*-carotene derived cleavage product regulates plastid development and PLB formation independent of GUN-mediated signalling

Due to their hydrophobicity and *cis*-configuration, *cis*-carotenes were hypothesised to function as a membrane-bound structural inhibitor of PLB formation during skotomorphogenesis (**Park et al., 2002**; **Cuttriss et al., 2007**). Herin, we experimentally demonstrate that *ccr2* generated a *cis*-carotene-derived cleavage product, as D15 chemical inhibition of CCD activity (**Figure 1—figure supplement 1A**) restored PLB formation (85%) in *ccr2* etioplasts (**Figure 6**). This is in agreement with evidence showing *cis*-carotenes are cleavable in vitro by CCD7 enzymatic activity (**Bruno et al., 2016**) and that CCD4 activity is necessary for generation of a *cis*-carotene derived apocarotenoid signal in *zds/clb5,* which affected leaf development (**Avendaño-Vázquez et al., 2014**). However, loss-of-function of *ccd1, ccd4, ccd7* and *ccd8* was not sufficient to restore plastid development and prevent leaf virescence in *ccr2* (**Figure 6—figure supplement 1**). So, we conclude that there must be some redundancy among two or more CCDs in generating a *ccr2* derived apocarotenoid signalling metabolite that controls plastid development.

Which *cis*-carotene is the precursor for the apocarotenoid signal? Treatment with NF restored PLB formation in *ccr2* etioplasts (**Cuttriss et al., 2007**) ruling out both phytoene and phytofluene as substrates for the generation of a cleavage product, since they accumulate following NF-mediated inhibition of PDS activity (**Figure 1—figure supplement 1A**). Herein we show that the *ziso* mutation restored PLB formation and cotyledon greening in *ccr2* ruling out tri-*cis*-ζ-carotene and revealing that di-*cis*-ζ-carotene, pro-neurosporene isomers and/or tetra-*cis*-lycopene are likely candidates (**Figure 4**). *ccr2 det1-154* displayed a substantial reduction in pro-neurosporene and tetra-*cis*-lycopene, and to a lesser extent di-*cis* ζ-carotene (**Figure 5—figure supplement 1**). Tetra-*cis*-lycopene accumulated in virescent leaves (evident as shown in transverse pale and green stripes that presumably resulted from impaired chloroplast development in leaf primordia cells that differentiated during dark periods) from the rice *zebra* mutant (**Han et al., 2012**). However, in the presence of D15 and hence absence of any enzymatic cleavage, only di-*cis*-ζ-carotene and pro-neurosporene accumulated, not tetra-*cis*-lycopene (**Figure 6**). Based on the evidence to date, we consider pro-neurosporene and perhaps di-*cis*-ζ-carotene are preferred substrate(s) for in vivo cleavage into a signalling metabolite.

Does the proposed apocarotenoid require GUN activity to regulate PLB formation and/or *PhANG* expression? Given that *gun1* etioplasts contain PLBs, then that aspect of the *ccr2* phenotype is not GUN-related (**Susek et al., 1993**; **Xu et al., 2016**). There were relatively few DE genes in common between *ccr2* etiolated seedlings and *gun* mutants or far red light block of greening seedlings treated with norflurazon (**Supplementary file 5**). None of the 25 revertant lines were in genic regions to which *GUN* genes are located. Norflurazon treatment of etiolated tissues does not affect PLB formation in wild type, but can restore PLB formation in *ccr2* (**Cuttriss et al., 2007**; **Xu et al., 2016**). Lincomycin treatment, on the other hand can suppress PLB formation in etiolated seedlings and unlike norflurazon, affects the phenotype of *pifq* mutant (quadruple mutant of *pif1 pif3 pif4 pif5*) seedlings grown in the dark. GUN1-facilitated retrograde signalling antagonized *PIF*-regulated gene expression and attenuated de-etiolation phenotypes triggered by lincomycin (**Martín et al., 2016**). Lincomycin can inhibit PLB formation in the *pifq* mutant, revealing that PIFs are not necessary for PLB formation (**Martín et al., 2016**). Previous research showed that *hy5*, *pif3* and *pifq* dark grown seedlings contain etioplasts with PLBs, albeit in some cases reduced in size (**Chang et al., 2008**; **Stephenson et al., 2009**; **Martín et al., 2016**). GUN1-dependent and independent signalling pathways were proposed to act upstream of HY5 to repress photomorphogenesis of cotyledons (**Ruckle et al., 2007**). Intriguingly, the *ccr2* generated *cis*-carotene derived cleavage product also regulated a distinct set of genes involved in a photomorphogenic-dependent pathway. The nature by which a *cis*-carotene derived cleavage product regulates PLB formation by transcriptionally enhancing *PORA* is quite distinct to that of GUN regulation of *PhANG* gene expression. Consequently, we deduce that the lack of a PLB in *ccr2* is neither a consequence of apocarotenoid

regulation of PIF3 or HY5, nor a lack of POR. As an alternative hypothesis we propose that the apocarotenoid signal and DET1 regulate an unknown factor necessary for PLB formation that is independent of POR abundance and GUN activity (*Figure 7E*).

## An apocarotenoid signal regulated *PIF3* and *HY5* transcript levels

Here we demonstrate that the *ccr2*-generated apocarotenoid acted in a retrograde manner to transcriptionally regulate *POR*, *PIF3* and *HY5* transcript levels in *ccr2* and *ccr2 det1-154* backgrounds (*Figures 7* and *8*). Of particular interest is how the abundances of all three transcript and protein levels were reverted in *ccr2 det1-154* to expected levels for *det1* mutants by treatment with D15. Similarly, D15 reverted *HY5* and *PIF3* transcript and protein levels back to WT levels in *ccr2*. The contra-regulation of the PIF3/HY5 ratio by D15 treatment was further contrasted by an even lower PIF3/HY5 ratio (due to the up-regulation of *HY5* and reduction in *PIF3* transcript levels compared to WT) in the *ccr2 ziso-155* mutant etiolated seedlings and leaves collected from plants grown under a shorter photoperiod (*Table 2*). While D15 has been shown to only impair substrate cleavage, the loss-of-function in ZISO can block substrate production in the dark (*Figure 4*) and limit biosynthesis of tri-*cis*-ζ-carotene and neurosporene in leaves from *ccr2 ziso-155* plants grown under a shorter photoperiod when photoisomerisation becomes rate-limited. Trace levels of *cis*-carotenes were detected in *det1* mutants (*Supplementary file 6*), Arabidopsis WT floral tissues (*Figure 2*), as well different tissues types from other species (*Alagoz et al., 2018*). Under favourable conditions of light, when activity of ZISO, CRTISO as well as photoisomerisation are not limited, the absence of the *cis*-carotene derived cleavage product enables the lowering of the PIF3/HY5 ratio to maintain *PhANG* expression in concert with chloroplast development. The broader genetic regulation of the *ccr2*-generated apocarotenoid signal and the role of light in controlling its abundance and/or mechanism of action will be able to be elucidated once the signal has been identified.

## An apocarotenoid signal acts in parallel with DET1 to regulate plastid development

*cis*-carotenes will hyper-accumulate in etiolated cotyledons and younger leaf tissues exposed to an extended dark period when CRTISO activity becomes rate-limited, such as in the absence of chromatin-modifying enzyme, SDG8. SDG8 is required for permissive expression of *CRTISO* in developing seedlings and shoot meristem (*Cazzonelli et al., 2009b*; *Cazzonelli et al., 2009a*; *Cazzonelli et al., 2010*) (*Figure 2*). *SDG8* transcript levels are developmentally regulated, increasing from low basal levels after germination and declining during the dark phase of the diurnal cycle (*Kim et al., 2005*). Here we linked a perturbation in *cis*-carotene accumulation in *ccr2 ziso-155* juvenile leaves grown under a shorter photoperiod to an enriched gene ontology in chromatin and DNA modifying processes, in particularly the repression of *DET1* gene expression (*Supplementary file 4* and *Table 2*). DET1 was required for *cis*-carotene biosynthesis in wild type tissues, as *det1* mutants accumulate phytoene, phytofluene and tri-*cis*-ζ-carotene (*Supplementary file 6*). Similarly, the down-regulation and/or loss-of-function in *det1* in tomato lines revealed an accumulation of phytoene and phytofluene in ripe fruits (*Enfissi et al., 2010*; *Talens et al., 2016*). Therefore, CRTISO and DET1 can control the accumulation of *cis*-carotenes and the apocarotenoid signal that regulates plastid development, a process that perhaps is fine-tuned with epigenetic and chromatin modifying processes that control light-mediated development.

Herein we revealed how *ccr2* and *det1* oppositely regulate the chlorophyll biosynthetic enzyme, POR at the transcriptional level during skotomorphogenesis (*Figure 7*). There are relatively few mutants published to date that do not produce a PLB in dark grown tissues and all, except *ccr2,* are due to reduced levels of *PORA* and/or PChlide. Arabidopsis mutants like *det1-1* and *cop1* mutants have less photoactive PChlide-F655 and higher total PChlide levels due to a reduction in POR that thereby blocks PLB formation (*Sperling et al., 1998*). Like *det1-1*, *det1-154* exhibits all the same phenotypes and indeed D15 treatment of *ccr2 det1-154* blocked PLB formation (*Chory et al., 1989*) (*Figure 5—figure supplement 1*; *Table 1*). While etioplasts in *ccr2* dark grown cotyledons do not make a PLB, the abundance of POR and PChlide levels are similar to wild type (*Figures 6* and *7*). Therefore, *ccr2* and *det1* control PLB formation via distinct, although perhaps mutually dependent signalling pathways, whereby the *cis*-carotene derived signal blocks the *det1*-mediated transcriptional repression of *PORA* gene expression. Can the *ccr2*-derived cleavage product directly

regulate DET1? This is unlikely for several reasons. First, *ccr2* and *ccr2 ziso-155* displayed closed cotyledons, an apical hook and normal hypocotyl length revealing that the *cis*-carotene derived cleavage metabolite does not activate photomorphogenesis (*Table 1*). Second, DET1 protein levels were relatively unchanged in WT, *ccr2*, *det1-154* and *ccr2 det1-154*, regardless of D15 chemical inhibition. Hence, the *cis*-carotene-derived apocarotenoid cleavage product can transcriptionally up-regulate POR levels in *det1*, thereby enabling PLB formation in etioplasts and chloroplast differentiation following de-etiolation.

*DET1* encodes a nuclear protein acting downstream from the phytochrome photoreceptors to negatively regulate light-driven seedling development and promote skotomorphogenesis (*Schroeder et al., 2002*). DET1 interacts with COP1 and the chromatin regulator DDB1, to limit the access of transcription factors to promoters and negatively regulate the expression of hundreds of genes via chromatin interactions (*Schroeder et al., 2002*; *Lau and Deng, 2012*). Light stimulates photomorphogenesis and the rapid down-regulation of *DET1* leading to a lower PIF3/HY5 protein ratio and the up-regulation of *PhANG* expression according to published results. Genetic mutations in *cop1* and *det*1 also lower the PIF3/HY5 ratio and activate *PhANG* expression (*Osterlund et al., 2000*; *Benvenuto et al., 2002*). Consistent with these findings, *ccr2 det1-154* etiolated and de-etiolated seedlings treated with D15 displayed higher HY5 and lower PIF3 protein levels, contrasting opposite to that of *ccr2* (*Figures 7* and *8*). The *cis*-carotene derived cleavage metabolite can transcriptionally antagonise the DET1 mediated post-transcriptional regulation of HY5 and PIF3. In conclusion, we deduce that the unknown apocarotenoid retrograde signal acts at the transcriptional level in parallel with the negative regulator DET1, to control POR, PIF3 and HY5 and thus regulate etioplast development during skotomorphogenesis and chloroplast development under extended periods of darkness (*Figure 9*).

## Materials and methods

### Mutants used in this study

All germplasms are in the *Arabidopsis thaliana* ecotype Columbia (Col-0) background except where otherwise indicated. Germplasm used in this study include; *ziso*#11C (*zic1-3*: Salk_136385), *ziso*#12D (*zic1-6*; Salk_057915C), *ziso*#13A (*zic1-4*; CS859876), *ccr2-1/crtiso* (*Park et al., 2002*), *ccr1-1/sdg8* (*Cazzonelli et al., 2009b*), *lut2-1* (*Pogson et al., 1996*), *ccd1-1* (SAIL_390_C01), *ccd4* (Salk_097984 c), *max3-9/ccd7* (*Stirnberg et al., 2002*), *max4-1/ccd8* (*Sorefan et al., 2003*), *aba1-3* (*Koornneef et al., 1982*), *det1-1* (CS6158). *ziso-155*, *ccr2 ziso-155*, *ccr2 det1-154* and *det1-154* were generated in this study.

A forward genetics and second site revertant screen was accomplished by mutagenizing seeds in ethyl-methane sulfonate (EMS) as previously described (*Weigel and Glazebrook, 2006*). EMS treated seeds were sown in soil, plants grown and seeds collected from pools of 5–10 $M_1$ plants. Approximately 40,000 $M_2$ seedlings from 30 stocks of pooled $M_1$ seeds were screened for the emergence of green juvenile rosette leaves that were not virescent when grown under a 10 hr photoperiod.

### Plant growth conditions and treatments

For soil grown plants, seeds were sown on DEBCO seed raising mixture and stratified for 3 d at 4°C in the dark, prior to transferring to an environmentally controlled growth chamber set to 21°C and illuminated by approximately 120 $\mu mol.m^{-2}.sec^{-1}$ of fluorescent lighting. Unless otherwise stated, plants were grown in a 16 hr photoperiod. Photoperiod shift assays were performed by shifting 2–3 week old plants grown under a 16 hr photoperiod to an 8 hr photoperiod for one week and newly emerged immature leaves were scored as displaying either a yellow leaf (YL) or green leaf (GL) phenotype, reflecting either impaired or normal plastid development respectively.

For media grown seedlings, Arabidopsis seeds were sterilized for 3 hr under chlorine gas in a sealed container, followed by washing seeds once with 70% ethanol and three times with sterilized water. Seeds were sown onto Murashige and Skoog (MS) media (Caisson Labs; MSP01) containing 0.5% phytagel (Sigma) and half-strength of Gamborg's vitamin solution 1000X (Sigma Aldrich) followed by stratification for 2 d (4°C in dark) to synchronise germination. Inhibition of carotenoid cleavage dioxygenase (CCD) enzyme activity was achieved by adding D15 (aryl-C3N hydroxamic acid)

dissolved in ethanol to a final optimal concentration of 100 µM as previously described (*Van Norman et al., 2014*). Etiolation experiments involved growing seedlings in petri dishes containing MS media and incubating them in dark at 21°C for 7 d, after which cotyledons were harvested under a dim green LED light. For de-etiolation and greening experiments, Arabidopsis seeds were stratified for 2 d and germinated in the dark at 21°C for 4 d. Seedlings were then exposed to constant light (~80 µmol.m$^{-2}$.sec$^{-1}$, metal-halide lamp) for 72 hr at 21°C. Cotyledon tissues were harvested at 24 hr intervals for chlorophyll quantification.

## Plasmid construction

pEARLEY::ZISO-OE and pEARLEY::DET1-OE binary vectors were designed to overexpress *ZISO* and *DET1* cDNA fragments, respectively. Both genes were regulated by the constitutive CaMV35S promoter. Full length cDNA coding regions were chemically synthesised (Thermo Fisher Scientific) and cloned into the intermediate vector pDONR221. Next, using gateway homologous recombination, the cDNA fragments were cloned into pEarleyGate100 vector as per Gateway Technology manufacturer's instructions (Thermo Fisher Scientific). Vector construction was confirmed by restriction digestion and Sanger sequencing.

## Generation of transgenic plants

The *ccr2 ziso-155* and *ccr2 det1-154* EMS generated mutant lines were transformed by dipping Arabidopsis flowers with Agrobacteria harbouring pEARLEY::ZISO-OE or pEARLEY::DET1-OE binary vectors to generate *ccr2 ziso-155*::ZISO-OE and *ccr2 det1*[154]::DET1-OE transgenic lines, respectively. At least 10 independent transgenic lines were generated by spraying seedlings grown on soil with 50 mg/L of glufosinate-ammonium salt (Basta herbicide).

## Chlorophyll pigment quantification

Total chlorophyll was measured as described previously (*Porra et al., 1989*) with minor modifications. Briefly, 20 seedlings from each sample were frozen and ground to fine powder using a TissueLyser (Qiagen). Homogenised tissue was rigorously suspended in 300 µL of extraction buffer (80% acetone and 2.5 mM NaH$_2$PO$_4$, pH 7.4), incubated at 4°C in dark for 15 min and centrifuged at 20,000 g for 10 min. Two hundred and fifty microliters of supernatant was transferred to a NUNC 96-well plate (Thermo Fisher Scientific) and measurements of A647, A664 and A750 were obtained using an iMark Microplate Absorbance Reader (Thermo Fisher Scientific). Total chlorophyll in each extract was determined using the following equation modified from *Porra (2002)*: (Chl a + Chl b) (µg) = (17.76 × (A647-A750) + 7.34 × (A664-A750))×0.895 × 0.25.

## Carotenoid pigment analysis

Pigment extraction and HPLC-based separation was performed as previously described (*Cuttriss et al., 2007*; *Dhami et al., 2018*; *Alagoz et al., 2020*). Reverse phase HPLC (Agilent 1200 Series) was performed using either the GraceSmart-C18 (4 µm, 4.6 × 250 mm column; Alltech) or Allsphere-C18 (OD2 Column 5 µm, 4.6 × 250; Grace Davison) and/or YMC-C30 (250 × 4.6 mm, S-5µm) columns. The C18 columns were used to quantify β-carotene, xanthophylls and generate *cis*-carotene chromatograms, while the C30 column improved *cis*-carotene separation and absolute quantification. Carotenoids and chlorophylls were identified based upon retention time relative to known standards and their light emission absorbance spectra at 440 nm (chlorophyll, β-carotene, xanthophylls, pro-neurosporene, tetra-cis-lycopene), 400 nm (ζ-carotenes), 340 nm (phytofluene) and 286 nm (phytoene). Absolute quantification of xanthophyll pigments was performed as previously described (*Pogson et al., 1996*). Quantification of *cis*-carotenes was performed by using their molar extinction coefficient and molecular weight to derive peak area in terms of micrograms (µg) per gram fresh weight (gfw) (*Britton, 1995*).

## Transmission Electron Microscopy (TEM)

Cotyledons from 5-d-old etiolated seedlings were harvested in dim-green safe light and fixed overnight in primary fixation buffer (2.5% Glutaraldehyde and 4% paraformaldehyde in 0.1 M phosphate buffer pH 7.2) under vacuum, post-fixed in 1% osmium tetroxide for 1 hr, followed by an ethanol series: 50%, 70%, 80%, 90%, 95% and 3 × 100% for 10 min each. After dehydration, samples were

incubated in epon araldite (resin): ethanol at 1: 2, 1: 1 and 2:1 for 30 min each, then 3 times in 100% resin for 2 hr. Samples were then transferred to fresh resin and hardened under nitrogen air at 60°C for 2 d, followed by sectioning of samples using Leica EM UC7 ultramicrotome (Wetzlar). Sections were placed on copper grids, stained with 5% uranyl acetate, washed thoroughly with distilled water, dried, and imaged with H7100FA transmission electron microscope (Hitachi) at 100 kV. For each of the dark-grown seedling samples, prolamellar bodies were counted from 12 fields on three grids, and data analysed using two-way ANOVA with post-hoc Tukey HSD.

## DNA-seq library construction, Sequencing and Bioinformatics Identification of SNPs

Genomic DNA (gDNA) was extracted using the DNeasy Plant Mini Kit (Qiagen). One microgram of gDNA was sheared using the M220 Focused-Ultrasonicator (Covaris) and libraries were prepared using NEBNext Ultra DNA Library Prep Kit (New England Biolabs) followed by size selection (~320 bp) using AMPure XP Beads (Beckman Coulter). Paired-end sequencing was performed using the Illumina HiSEQ1500. After sequencing, the raw reads were assessed for quality using the FastQC software (http://www.bioinformatics.babraham.ac.uk/ projects/fastqc/), and subjected to trimming of illumina adapters and filtering of low quality reads with AdapterRemoval programme (*Lindgreen, 2012*). The reads were mapped to the *Arabidopsis thaliana* (TAIR9) genome with BWA mapper (*Li and Durbin, 2009*). The resultant BWA alignment files were converted to sorted bam files using the samtools v0.1.18 package (*Li et al., 2009*) and were used as input for the subsequent SNP calling analyses. The SNPs were called and analysed further on both the parent and mutant lines using NGM pipeline (*Austin et al., 2011*) and SHOREmap (*Schneeberger et al., 2009*). For the NGM pipeline, SNPs were called using samtools (v0.1.16) as instructed and processed into '.emap' files using a script provided on the NGM website. The. emap files were uploaded to the NGM web-portal to assess SNPs with associated discordant chastity values. To identify mutant specific SNPs, SNPs from parental lines were filtered out and EMS-induced homozygous SNPs were defined based on the discordant chastity metric. For SHOREmap, the SHORE software (*Ossowski et al., 2008*) was used to align the reads (implementing BWA) and call the SNPs (*Hartwig et al., 2012*). SHOREmap backcross was then implemented to calculate mutant allele frequencies, filter out parent SNPs and define the EMS mutational changes. Where appropriate, custom scripts were used to identify mutant specific EMS SNPs, filter out parent SNPs and annotate the region of interest. The SNPs and InDels were localized based on the annotation of gene models provided by TAIR database (http://www.arabidopsis.org/). The polymorphisms in the gene region and other genome regions were annotated as genic and intergenic, respectively. The genic polymorphisms were classified as CDS (coding sequences), UTR (untranslated regions), introns and splice site junctions according to their localization. SNPs in the CDS were further separated into synonymous and non-synonymous amino substitution. The GO/PFAM annotation data were further used to functionally annotate each gene.

## RNA-seq library construction, Sequencing and Differential Gene Expression Analysis

Total RNA was extracted from Arabidopsis leaf tissues grown under an 8 hr photoperiod or cotyledons from etiolated seedlings grown in dark for 7 d by TRIzol (Thermo Fisher Scientific) followed by DNase treatment at 37°C for 30 min. RNA was recovered using x1.8 Agencourt RNAClean XP magnetic beads (Beckman Coulter). RNA (1 µg) libraries were constructed using Illumina TruSeq Stranded mRNA Library Prep Kit (ROCHE) followed by bead size selection (~280 bp) using AMPure XP Beads and libraries sequenced using the Illumina HiSEQ2000. Fifteen million reads were obtained from sequencing each library and 21365 to 23840 mRNA transcripts were identified. Quality control was performed with FASTQC v.0.11.2. Adapters were removed using scythe v.0.991 (flags -p 0.01 for the prior), reads trimmed with sickle v.1.33 (flags q 20; quality threshold and -l 20 for minimum read length after trimming) and aligned to the Arabidopsis genome (TAIR10) using the subjunc v.1.4.6 aligner (-u and -H flags to report reads with a single, unambiguous mapping location) (*Liao et al., 2014*). The number of reads mapping per gene were summarised using feature Counts (v.1.4.6 with flags -s 2, -P and -c) to map reverse stranded and discard read pairs mapping to different chromosomes (*Liao et al., 2014*). Statistical testing for relative gene expression was performed in R using edgeR v.3.4.2 (*Robinson and Smyth, 2007*; *Robinson and Smyth, 2008*; *Robinson et al.,*

*2010*; *Robinson and Oshlack, 2010*; *McCarthy et al., 2012*), Voom (*Law et al., 2014*) in the limma package 3.20.1 (*Smyth, 2004*; *Smyth, 2005*). Transcripts were considered differentially expressed when a fold change >2 and FDR adjusted p<0.05. The bioinformatics analysis pipeline from fastq to summarised counts per gene is available at https://github.com/pedrocrisp/NGS-pipelines. RNAseq data sets was deposited into a permanent public repository with open access (https://www.ncbi.nlm.nih.gov/sra/PRJNA498324).

## Protein extraction and western blot analysis

For protein extraction, fifty to one hundred milligrams of etiolated Arabidopsis cotyledons (7-d-old) were harvested under dim-green safe light and ground to fine powder. Total protein was extracted using a TCA-acetone protocol (*Méchin et al., 2007*) with minor modification and pellets were suspended in 100 μL – 200 μL solubilization buffer. The concentration of total protein was measured using Bradford reagent (Bio-Rad) and adjusted to 2 μg/μL. A serial dilution was used to determine western blot sensitivity for each antibody and determine the optimal concentration for quantification. Five micrograms of total protein run on a gel was transferred to a PVDF membrane (Bio-Rad) and incubated with anti-POR polyclonal antibody (Agrisera Antibodies AS05067, 1:2000), anti-HY5 antibody (Agrisera Antibodies AS121867, 1:1000), anti-LHCB2 antibody (Agrisera Antibodies AS01003, 1:1000, gift from Dr Spencer Whitney) or anti-PIF3 polyclonal antibody (Agrisera Antibodies AS163954, 1:2000) for 2 hr. To examine DET1 protein levels, 10 μg of total protein was loaded to the gel and anti-DET1 polyclonal antibody (Agrisera Antibodies AS153082) was used at a 1:1000 dilution. Membranes were washed and incubated with HRP-conjugated Goat anti-Rabbit IgG (Agrisera Antibodies AS09602, 1:5000) for 90 min or for PIF3 with HRP-conjugated Rabbit anti-Goat IgG (Agrisera Antibodies AS09605, 1:5000) for 90 min. Membranes were re-probed using anti-Actin polyclonal antibody (Agrisera Antibodies AS132640, 1:3000) and HRP-conjugated Goat anti-Rabbit IgG (Agrisera Antibodies AS09602, 1:5000) for internal protein normalisation.

## Protochlorophyllide quantification

Protochlorophyllides (Pchlides) were extracted and measured using published methods (*Kolossov and Rebeiz, 2003*) with modifications. Around 100 mg of etiolated Arabidopsis seedlings (7-d-old) were harvested under dim-green safe light, frozen and ground to fine powder. Two milliliters of 80% ice-cold acetone was added to each sample and the mixture was briefly homogenized. After centrifugation at 18,000 g for 10 min at 1 $^{0}$C, supernatant was split to 2 × 1 mL for Pchlides and protein extraction. Fully esterified tetrapyrroles were extracted from the acetone extracts with equal volume followed by 1/3 vol of hexane. Pchlides remained in the hexane-extracted acetone residue were used for fluorescence measurement with a TECAN M1000PRO plate reader (Tecan Group) and net fluorescence were determined as previously described (*Rebeiz et al., 1975*). Protein extraction was performed using 80% acetone and 10% TCA; protein concentration was used to normalize the net fluorescence of Pchlides.

## Real-Time PCR analysis

The total RNA was extracted using Spectrum Plant Total RNA kit as per manufacturer's protocol (Sigma-Aldrich). The qRT-PCR was performed with mixture of 2 μL of primer mix (2 μM from each F and R primer), 1 μL 1/10 diluted cDNA template, 5 μL LightCycler 480 SYBR Green I Master mix and distilled water up to a total volume of 10 μL. Relative transcript abundance was quantified using LightCycler 480 as per instructions (Roche). For each sample, three technical replicates for each of three biological replicates were tested. The relative gene expression levels were calculated by using relative quantification (Target Eff Ct(Wt-target)/Reference Eff Ct(Wt-target)) and fit point analysis (*Pfaffl, 2001*). Protein Phosphatase 2A (At1g13320) was used as housekeeper reference control for all experiments (*Czechowski et al., 2005*) (*Czechowski et al., 2005*). All primer sequences are listed in *Supplementary file 7*. Statistical analysis was performed using Two-Way ANOVA.

## Acknowledgements

We especially thank Rishi Aryal (confirmed the splicing of *det1-154* with YA) and Peter Crisp (assisted XH with the RNA bioinformatics analysis) for their technical assistance. Many thanks to Arun Yadav, Shelly Verma, William Walker, Michelle Nairn, Sam Perotti, Jacinta Watkins and Kai Chan for their

assistance in maintaining plants, crossing mutant germplasm and performing HPLC. We thank Philip Benfey for providing the D15 chemical inhibitor of CCD activity. Next generation sequencing was performed at the Biomolecular Resource Facility (ANU). This work was supported by Grant CE140100008 (BJP) and DP130102593 (CIC).

## Additional information

### Funding

| Funder | Grant reference number | Author |
| --- | --- | --- |
| Centre of Excellence in Plant Energy Biology, Australian Research Council | CE140100008 | Barry J Pogson |
| Australian Research Council | DP130102593 | Christopher I Cazzonelli Barry J Pogson |

The funders had no role in study design, data collection and interpretation, or the decision to submit the work for publication.

### Author contributions

Christopher I Cazzonelli, Conceptualization, Resources, Data curation, Formal analysis, Supervision, Funding acquisition, Validation, Investigation, Visualization, Methodology, Writing - original draft, Project administration, Writing - review and editing, Supervised XH, JR, ND and YA, Prepared figures and tables and performed the majority of experiments; Xin Hou, Data curation, Software, Formal analysis, Validation, Investigation, Methodology, Writing - original draft, Writing - review and editing, Prepared figures and tables and performed the majority of experiments; Yagiz Alagoz, Formal analysis, Validation, Investigation, Methodology, Writing - review and editing, Contributed to Figures 5, 6, 7, 8, Table 1, Supplementary File 6 and Figure 5-figure supplement 1; John Rivers, Investigation, Methodology, Writing - review and editing, Produced Figure 6-figure supplement 1; Namraj Dhami, Investigation, Methodology, Writing - review and editing, Contributed to Table 1, Supplementary File 6 and Figure 5-figure supplement 1; Jiwon Lee, Investigation, Methodology, Contributed expertise in TEM; Shashikanth Marri, Data curation, Software, Formal analysis, Methodology, Writing - review and editing, Performed the DNA bioinformatics analysis; Barry J Pogson, Conceptualization, Resources, Formal analysis, Supervision, Funding acquisition, Validation, Investigation, Visualization, Methodology, Project administration, Writing - review and editing, Supervised XH, JR and ND

### Author ORCIDs

Christopher I Cazzonelli (iD) https://orcid.org/0000-0003-3096-3193
Xin Hou (iD) https://orcid.org/0000-0003-4625-1692
Yagiz Alagoz (iD) https://orcid.org/0000-0001-5081-6874
John Rivers (iD) https://orcid.org/0000-0002-7500-5637
Namraj Dhami (iD) https://orcid.org/0000-0002-6014-3500
Barry J Pogson (iD) https://orcid.org/0000-0003-1869-2423

### Decision letter and Author response

Decision letter https://doi.org/10.7554/eLife.45310.sa1
Author response https://doi.org/10.7554/eLife.45310.sa2

## Additional files

### Supplementary files

• Supplementary file 1. Immature *ccr2* tissues have an altered *cis*-carotene and xanthophyll composition.
• Supplementary file 2. D15 and *ziso* restore PLB formation in *ccr2* etiolated cotyledons.

- Supplementary file 3. Transcriptomic analysis of WT, *ccr2* and *ccr2 ziso-155* etiolated tissues.
- Supplementary file 4. Transcriptome analysis of WT, *ccr2* and *ccr2 ziso-155* immature leaf tissues.
- Supplementary file 5. Significantly expressed genes regulated in *ccr2* and contra-regulated *ccr2 ziso-155* that are common to both etiolated and immature leaf tissues.
- Supplementary file 6. *det1* reduced carotenoids and caused *cis*-carotenes to accumulate in leaves and etiolated tissues.
- Supplementary file 7. Primer sequences used for qRT-PCR and *ccr2 det154* characterisation.
- Transparent reporting form

## Data availability

Data availability information is outlined in the methods and materials, figure legends and/or results sections. Supplementary files 3, 4, and 5 refer to additional files describing transcriptomics data (RNAseq). The bioinformatics analysis pipeline from fastq to summarised counts per gene is available at https://github.com/pedrocrisp/NGS-pipelines. RNAseq data sets were deposited into a permanent public repository with open access (https://www.ncbi.nlm.nih.gov/sra/PRJNA498324).

The following dataset was generated:

| Author(s) | Year | Dataset title | Dataset URL | Database and Identifier |
|---|---|---|---|---|
| Cazzonelli CI, Hou X, Pogson BJ | 2018 | A cis-carotene derived cleavage product acts downstream of deetiolated 1 to control protochlorophyllide oxidoreductase and prolamellar body formation | https://www.ncbi.nlm.nih.gov/sra/PRJNA498324 | NCBI Sequence Read Archive, PRJNA498324 |

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
