## [Decision Letter]

**Acceptance summary:**

We and the reviewers consider your work as a substantial mechanistic breakthrough, revealing a novel apocarotenoid retrograde signal. We therefore thank you for choosing *eLife* to publish this innovative manuscript.

**Decision letter after peer review:**

Thank you for submitting your article "A *cis*-carotene derived apocarotenoid regulates etioplast and chloroplast development" for consideration by *eLife*. Your article has been reviewed by three peer reviewers, and the evaluation has been overseen by Jürgen Kleine-Vehn as the Reviewing Editor and Christian Hardtke as the Senior Editor. The following individual involved in review of your submission has agreed to reveal their identity: Enrique Lopez Juez (Reviewer #2).

The reviewers have discussed the reviews with one another and the Reviewing Editor has drafted this decision to help you prepare a revised submission.

Your extensive work links carotenoid biosynthesis and plastid development, revealing that an apocarotenoid signaling molecule acts in plastid formation. The reviewers were largely positive and agreed that your work is in principle suitable for *eLife*. However, the reviewers also raised some substantial concerns (see detailed reviewer comments below), which need to be addressed.

Essential revisions:

I) Additional controls (*ccr2* and *det1-154* mutants) are needed for Figure 7.

II) The reviewers ask you to address the light/dark dependency of the observed effects in Figure 7.

III) The transcriptomic data analysis and presentation should be improved. The reviewers would also like to see how your data set compares to published data sets (e.g. Page et al., 2017).

IV) Apparent discrepancy between data in main and supplementary figure need to be explained and clarified.

V) Interpretation/discussion of the data needs to be thoroughly improved in order to address the indicated inconsistencies highlighted by the reviewers (see below).

Reviewer #1:

This manuscript describes a role for carotenoid cleavage products as regulators of plastid biogenesis and that this regulation is mediated, at least in part, via a signal that leaves the chloroplast to control levels of key transcription factors. Clear demonstration that another apocarotenoid signal is a retrograde regulator of plastid development would be a significant advance in our understanding of this important process. This is a thorough study and the quality of the presented data is very high. The interpretation of the experiments with respect to an apocarotenoid signal regulating plastid development is sound, but is more difficult to follow when discussing the mechanism for this. This is partly because the paper is not clearly written in places. The Discussion in particular left me less clear on what is being proposed than before I read it. The description of the gene expression studies was also difficult to follow with lots of different experiments conducted and the data only shown in some poorly explained supplementary files. The key result seems to be changes in expression of nuclear genes important in plastid development in Supplementary file 5 and Supplementary Table 6 (the data are duplicated). More could have been made of these data (table or figure in main paper) and it would have been nice to see some of these results confirmed by qPCR as the effect of the *det1-154* mutation and D15 could then be included.

Overall, although it is an interesting story, there are quite a few points I'm still confused about. Firstly, the role of carotenoid cleavage products in signalling. The *ccr2* mutant shows changes in nuclear gene expression that are reversed by the ziso mutation, but the absence of an effect of D15 leads to the conclusion that *cis*-carotenoid cleavage does not directly affect transcriptional regulation of these genes (subsection “A *cis*-carotene cleavage product acts downstream of DET1 to post-transcriptionally regulate protein levels”, first paragraph). So are we talking about two different signals, one a *cis*-carotene and one a *cis*-carotene cleavage product? The data in Figure 7A that addresses this point are very incomplete and need to include the *ccr2* and *det1-154* mutants. In addition, the PIF3 and HY5 genes show the opposite regulation compared to WT in Supplementary Table 6, but in this figure they are regulated similarly (though of course a direct comparison is not possible). What is the explanation for this?

It is also unclear why, when there are major changes in transcript abundance for PIF3 and HY5 in *ccr2*, it was concluded so strongly that regulation of the PIF3 and HY5 protein levels was post-transcriptional. The regulation of PIF3 and HY5 transcript abundance in *ccr2* with and without D15 needs to be completed before this can be concluded. Moreover, it is concluded (subsection “An apocarotenoid post-transcriptionally regulates PIF3 and HY5 protein levels”) that the lack of a PLB in *ccr2* is not due to the reduction in PIF3, but other PIFs (especially PIF1) have not been tested and multiple *pif* mutants do lack PLBs even if *pif3* alone doesn't.

The relationship with *det1* is quite difficult to understand. *det1* mutants lack a PLB and therefore the rescue of PLBs in *ccr2 det1-154* is extremely surprising in my view. A rescue in nuclear gene expression might account for it in part, but, although expected, this is not tested properly. Finally, there is also confusion in the related discussion of POR. Even in the Abstract it is stated that "*ccr2* has no PLB and normal POR" as well as that "the apocarotenoid signal and *det1* complemented each other to restore POR levels and PLB formation". If the *ccr2* phenotype is mediated by the production of apocarotenoids it is hard to reconcile these two statements. The statement in the second paragraph of the subsection “An apocarotenoid signal regulates skotomorphogenesis and plastid biogenesis in parallel to DET1” also cannot be true as it seems quite clear that the level of POR is not controlling PLB formation in this system (as opposed to the experiments of Armstrong et al. in which POR levels were modulated). The explanation later in the aforementioned subsection for *det1* rescue of *ccr2* doesn't make sense to me either. If the *cis*-carotene cleavage product is up-regulating POR and enhancing Pchlide levels (no evidence of this) to enable PLB formation why does *ccr2* not have a PLB? The *det1* mutant would only counter the POR effect.

In summary, this is a fascinating system that has the potential to reveal some important regulatory pathways in the control of plastid development, but some of the logical inconsistencies highlighted above need to be ironed out to make this story more coherent to the reader.

*Reviewer #2:*

This manuscript follows and dramatically expands on two sets of intriguing previous observations of the phenotypes of two carotenoid biosynthesis mutants, the current *ccr2*, defective in the enzyme CRITSO, as well as clbs5, defective in ZDS, the preceding enzyme in the synthesis of (all-trans) lycopene. According to the evidence obtained from both mutants, the authors propose an apocarotenoid signal (ACS) is derived from *cis*-carotenoids, which accumulate in both mutants. Such a signal has a range of developmental impacts, impairing the normal course of cellular differentiation and morphogenesis during leaf development in the light, It also causes a complex alteration of plastid development from dark (etioplast) to light (chloroplast), which in some respects prevents etioplast development, but in others enhances the characteristics which differentiate etioplasts from chloroplasts, and so may play a fundamental role in skotomorphogenesis.

The current manuscript includes an extraordinarily large body of observations which, in essence, are as follows:

- The Arabidopsis *ccr2* mutant does not have a visible phenotype under standard laboratory conditions. This is in contrast to tomato (tangerine) and rice (zebra2) mutants. *ccr2* is now shown to develop yellow early leaves, or a "virescent" phenotype, with leaf primordia or differentiating cells of developing leaves having underdeveloped chloroplast thylakoid membranes, only when grown under short day, long night photoperiods.

- The virescent phenotype is caused by accumulation of *cis*-carotenoids. Mutations in the previous enzyme in carotenoidd metablism, zeta-carotene isomerase, ZISO, or in the photomorphogenesis repressor DET1 revert the *ccr2* virescence phenotype. These are two of 25 revertant mutations of *ccr2* identified, and their identity is demonstrated by independent allele reversion and by wild type complementation. The *det1* revertant mutation indeed causes a deetiolated phenotype in the dark, something which is not shown by *ccr2*.

- The surprising, initially reported phenotype of *ccr2* is the absence of a prolamellar body (PLB) in etioplasts. The ziso revertant also restores the PLB in etioplasts.

- A whole transcriptome analysis of differentially expressed (DE) genes in *ccr2* or the *ccr2* ziso revertant shows that, although surprisingly more DE genes can be identified between the revertant and the wild type, the *ccr2* mutant is defective in photosynthesis associated nuclear genes and in some photomorphogenesis genes (HY5), while it shows elevated photomorphogenesis negative agents or regulators (PIF3, COP1, DET11). This is observed when examining etiolated seedlings (mutants and wild type) and also, surprisingly, young, virescent leaves. The gene expression reprogramming bears little resemblance to the changes which occur when seedlings are treated with the early carotenoid biosynthesis inhibitor norflurazon, which photobleaches chloroplasts.

- Both the absence of PLB and the virescence of *ccr2* can by suppressed by treatment with the pharmacological agent D15, a specific inhibitor of carotenoid cleavage dioxygenases, which act on *cis*-carotenoids and produce apocarotenoids.

- Treatment with D15 and observation of *ccr2* and the *ccr2 det1* revertant reveal a post-translational change in accumulation of PORA/B, a marker of etioplasts, PIF3, a skotomorphogenesis driver, and HY5, a photomorphogenesis driver, in a way consistent with the ACS (accumulating in *ccr2* but not the revertant and blocked by D15) promoting etioplast or skotomorphogenic development, and reducing photomorphogenic development. This action of the ACS appears additive, and most probably occurs in parallel to the action of DET1.

The volume and depth of the experiments in this study is extraordinary. While the identity of the ACS remains unknown, there is no doubt about its nature (a *cis*-carotene apocarotenoid or a result of its present) and impact. The role of the ACS in the development of etioplasts and in the skotomorphogenesis programme are particularly significant. There are, nevertheless, areas of the manuscript which need addressing:

1) The visible phenotype of CRITSO mutants, its basis and its relevance. This phenotype is variously described as "variegated" or "virescent", and is referred to as triggered by short days, a potential seasonal signal. This matters because the terminology suggests both mechanistic basis and fitness significance. A variegated mutation causes a stochastic failure in chloroplast development in some young, differentiating cells, but not others. As a result, some differentiated cell clusters/regions will show complete failure of greening, and others normal greening. This does not describe the phenotype of *ccr2*, where all young cells show impaired greening, and all completely differentiated ones appear normal. This is virescence. a kind of delayed chloroplast development. The rice CRITSO mutant, zebra2, shows transverse pale and green stripes, while variegated cereal mutants (like barley albostrians) show longitudinal stripes. The study of Han et al., 2012, showed that stripes of zebra2 disappeared when plants were grown under continuous light, even though the carotenoid composition appeared unaffected by photoperiod. In other words, the pale stripes resulted from periods in which chloroplast differentiation in leaf meristematic cells occurred in the dark. This study is cited but such observations are not. The observations with Arabidopsis *ccr2* are uncannily similar, the short photoperiod causing virescence being, more importantly, a long daily dark period. The most parsimonious explanation for the phenotype is the occurrence of a phenomenon or accumulation of a product in the dark, one which, upon light exposure, causes chloroplast development impairment, deceleration (arabidopsis and tomato) or arrest (rice).

2) The consequences of the action of ACS in the dark were described by Han et al. as production of reactive oxygen species, particularly singlet oxygen, upon light exposure of accumulated tetra-*cis*-lycopene. The authors of the current manuscript rightly highlight the differences between the transcriptome programme of *ccr2* seedlings or virescent leaves and those related to the "genomes uncoupled" regulation of photosynthesis associated nuclear genes. It has been shown (Page et al., 2017) that the repression of such genes by the "FR block of greening" treatment, which in fact impairs the correct etioplast-to-chloroplast transition, is associated to singlet oxygen damage. The source of the damaging oxygen species in this case is accumulated protochlorophyllide in the absence of its reductase, PORA (and of a prolamellar body). A comparison by the authors of the programme in *ccr2* and Page data could be revealing as to the root of the *ccr2* phenotype.

3) Particularly if the above turns out to be correct, one may not need to invoke a change in apocarotenoids as the cause for the reversion of *ccr2* by *det1*. Instead, *det1* would prevent the occurrence of a skotomorphogenic programme during the daily, extended dark period. Consequently, no photodynamic reagent (unbound protochlorophyllide?), source of singlet oxygen, or otherwise chloroplast repressive signal would accumulate during such periods.

4) The more parsimonious interpretation above does not diminish the importance of the ACS in the development of etioplasts or pre-chloroplasts. If could, instead, throw a new angle on it and emphasise its importance. It appears that the modified skotomorphogenic and the virescent stages of *ccr2* show commonalities in transcriptome programmes, and in the ratio of key transcription factor drivers of skoto- or photo-morphogenic stages (high and low PIF3/HY5 ratios respectively). As the authors note, this could place the still unidentified ACS signal in a central role in skotomorphogenesis.

5) Some of the protein blots in Figure 7 could have had separate control genotypes or conditions. Some description of exposures would help. See below.

6) The entire manuscript text uses a degree of confidence in the enzymatic activities and their responsible genes which the reviewer, admittedly not a specialist, wonders whether it can be justified. Early carotenoid biosynthesis genes are strongly seedling lethal. Losses of many later genes cause profile differences but relatively healthy development, which would be consistent with a combination of enzyme- and non-enzyme-mediated activities, or with redundancy in gene function, or both. This is referred in relation to the phenotypes of individual ccd mutants, but one wonders whether similar caution should be used throughout.

*Reviewer #3:*

This manuscript links carotenoid biosynthesis and plastid development with specific events in seedling and leaf development occurring in light and dark environments. In my view this manuscript presents a well done, detailed analysis of the links between these processes from carotenoid content to plastid morphology to gene expression and protein accumulation. I found it well written with thoughtfully constructed figures and supplementary materials. The finding that an apocarotenoid signaling molecule derived from linear carotenoids acts in plastid formation and development and skotomorphogenesis is a highly interesting finding and will have a broad impact from cellular to developmental biology to carotenoid metabolism onto environmental response. I have no substantive concerns and congratulate the authors on this well done manuscript.

[Editors' note: further revisions were suggested prior to acceptance, as described below.]

Thank you for resubmitting your work entitled "A *cis*-carotene derived apocarotenoid regulates etioplast and chloroplast development" for further consideration by *eLife*. Your revised article has been evaluated by Christian Hardtke (Senior Editor) and a Reviewing Editor.

The manuscript has been improved but there are some remaining issues that need to be addressed before acceptance, as outlined below:

The reviewers request more precision in describing "FR-block of greening". During the review consultation, the reviewers also discussed that the term "leaf meristematic cells" could be misleading and that the term "differentiating cells" would be more suitable.

Reviewer #2:

Cazonelli and collaborators have modified this manuscript in the following key ways:

1) The previous observations of *ccr2*-like mutants in other species, including the zebra mutants, are much more completely and carefully described. This includes the plastid-impairing impact of extended dark periods, which presumably led to their experimentation in the Arabidopsis mutant. This is accompanied with a more careful description of the enzyme- and non-enzyme-mediated conversion steps in carotenoid metabolism. While the authors do not emphasise these changes, they provide a key piece of evidence for the argument of the existence of the unidentified apocarotenoid signal (ACS) and its repressive impact on chloroplast development.

2) The global gene expression analysis of differentially expressed genes in etiolated seedlings and young leaves of the *ccr2* and *ccr2* ziso mutants has been more thoroughly carried out, incorporating it into the main text and adding comparisons to the effect of norflurazon treatments and "FR-block of greening".

3) The experimentation providing confusing data in the previous Figure 7, which lacked controls and also description of conditions, has been completely repeated. The results are difficult to compare with the previous, as those lacked a description of conditions, but appear rather different, and now much more in keeping with published data where relevant (on PIF3 and HY5). Those data now much more clearly support the authors' interpretation of the role the ACS.

4) The text and the model figures are partly rewritten and clearly improved.

Overall the changes make a very positive difference to the manuscript, which now provides clearer and more compelling evidence on the possible nature and the action of the ACS.

Particularly noteworthy is the fact that the authors refer to a role of light in the non-enzymatic isomerisation of tetra-*cis* lycopene, an alternative to the enzymatic step (CRTISO) lost in the *ccr2* mutant (see 1 above). This, combined with the lack of overlap with the singlet-oxygen regulated genes identified in the study of Page et al., 2017, strongly support a role for the ACS as opposed to the generation of a photodynamic molecule during the dark periods to explain the mutant phenotype. For this they should be congratulated, and this version of the manuscript is a very real improvement.

However the authors, in this reviewer's reading, describe that study of Page et al., and the impact of their experimental intervention, incorrectly. This is important because it obscures the relevance of the comparison and the importance of the findings. The so-called "FR-block of greening" cannot be described as "the action of FR light" or "a FR treatment". Rather it is the action of WHITE light on seedlings which had PREVIOUSLY been deetiolated under FR light. Such pretreatment results in a repression of PORA activity, while synthesis of Protochlorophyllide continues without a concomitant conversion into chlorophyllide. As a consequence, subsequent exposure to white light leads to absorption by Pchlide and generation of singlet oxygen, with lethal consequences for the seedlings overall. It was not impossible to conceive before that the extended dark periods resulted in the establishment of a similar condition in the *ccr2* mutant. The authors have now both ruled this out and provided an alternative explanation for the "corrective" effect of reducing dark periods. This issue, addressed at multiple points in the manuscript's text, needs redressing.

Reviewer #3:

This revision is addresses my previous comments and is well done resulting in an improved manuscript overall. Additionally, I believe the authors have adequately addressed the more substantial comments of the other two reviewers.

---

## [Author Response]

Essential revisions:I) Additional controls (ccr2 and det1-154 mutants) are needed for Figure 7.

Done. Additional mutants (*ccr2* and *det1-154*) along with WT and *ccr2 det1-154* (+/- D15) were included as requested. Experiments (qPCR and westerns) related to Figure 7 were repeated using new tissue samples. We further address specific reviewer comments below.

II) The reviewers ask you to address the light/dark dependency of the observed effects in Figure 7.

Done. Thank you for pointing out this discrepancy in the previous Figure 7 that was missed during review of the Materials and methods and figure legend. This has now been resolved by creating separate figures to avoid any confusion in how light vs. dark generated qPCR as well as western blot data. The new Figure 7 and 8 reveal dark (skotomorphogenesis) and light (photomorphogenesis) generated data, respectively. We further address specific reviewer comments below.

III) The transcriptomic data analysis and presentation should be improved. The reviewers would also like to see how your data set compares to published data sets (e.g. Page et al., 2017).

Done. This has provided additional insights into the revised manuscript. We updated the previous version of Supplementary Table 6 and moved this into the main body of paper, now presented as Table 2. We compared our DE gene expression data sets with Page et al., 2017. In brief, we did not find an overlap in differential gene expression between *ccr2* and Far Red light treatment of de-etiolated seedlings, and none of the contra-regulated in our study were responsive to Far Red light treatment as reported by Page et al., 2017. We have updated the Results and Discussion to highlight that an apocarotenoid (not singlet oxygen) is the signal responsible for regulating gene expression, PLB formation and plastid development. We addressed this further under reviewer #2 comments.

IV) Apparent discrepancy between data in main and supplementary figure need to be explained and clarified.

Done. This has been clarified by additional experimentation, enabling the discrepancy to be resolved. See comments addressing this concern in response to specific reviewers.

V) Interpretation/discussion of the data needs to be thoroughly improved in order to address the indicated inconsistencies highlighted by the reviewers (see below).

Done. We agree with the request and have substantial revised our Discussion and model to make the findings clearer and the impact for our understanding of skoto- and photo-morphogenesis clearer and more insightful.

In summary, we have added new data and included additional controls (Figures 7 and 8) as requested. We have now simplified our models for mechanisms of ACS regulation during skotomorphogenesis (Figure 7), photomorphogenesis (Figure 8) and under extended periods of darkness/shorter photoperiod (Figure 9). We have made better inclusion of our transcriptomics data by including a table in the manuscript, as well as comparing to the additional data set suggested by a reviewer. We substantially revised the manuscript to increase readability and to better highlight the most significant findings. We have taken actions to amend all changes suggested by each reviewer and provided responses to any queries. We thank all the reviewers for their insightful and constructive suggestions, as well as taking time to reveal necessary amendments, which as a direct result has now advanced the quality of our manuscript.

Reviewer #1:[…] The interpretation of the experiments with respect to an apocarotenoid signal regulating plastid development is sound, but is more difficult to follow when discussing the mechanism for this. This is partly because the paper is not clearly written in places. The Discussion in particular left me less clear on what is being proposed than before I read it. The description of the gene expression studies was also difficult to follow with lots of different experiments conducted and the data only shown in some poorly explained supplementary files. The key result seems to be changes in expression of nuclear genes important in plastid development in Supplementary file 5 and Supplementary table 6 (the data are duplicated). More could have been made of these data (table or figure in main paper) and it would have been nice to see some of these results confirmed by qPCR as the effect of the det1-154 mutation and D15 could then be included.

Done. We agree with the comments. The additional work we have done in the intervening months has provided more clarity and certainty about the findings enabling a much clearer, concise and insightful Discussion to be written. In addition, Supplementary Table 6 has been moved to Table 2 in main paper. The expression of *LHCB2.1*, one of the most well studied *PhANGs*, was confirmed by qPCR and western analysis, shown in Figure 8.

Overall, although it is an interesting story, there are quite a few points I'm still confused about. Firstly, the role of carotenoid cleavage products in signalling. The ccr2 mutant shows changes in nuclear gene expression that are reversed by the ziso mutation, but the absence of an effect of D15 leads to the conclusion that cis-carotenoid cleavage does not directly affect transcriptional regulation of these genes (subsection “A cis-carotene cleavage product acts downstream of DET1 to post-transcriptionally regulate protein levels”, first paragraph). So are we talking about two different signals, one a cis-carotene and one a cis-carotene cleavage product?

Thanks for this question. There is only one *cis*-carotene derived apocarotenoid (likely a cleavage product from neurosporene) signal (ACS) in this publication. We think this is a reflection of the confusion from the first version of the manuscript regarding to Figure 7. Our new version of this manuscript clarified how ACS transcriptionally regulates HY5/PIF3, POR and LHCB expression and that D15 contra-regulated these differential gene expression patterns.

The data in Figure 7A that addresses this point are very incomplete and need to include the ccr2 and det1-154 mutants.

Done. We agree and have provided a complete data set in the revised manuscript to address the reviewer’s concerns accordingly.

In addition, the PIF3 and HY5 genes show the opposite regulation compared to WT in Supplementary Table 6, but in this figure they are regulated similarly (though of course a direct comparison is not possible). What is the explanation for this?

We have repeated etiolated and de-etiolated experiments using all germplasm and examined transcript and protein levels for PORA, PIF3 and HY5 (See newly added data in Figure 7A and 8A). The trend in protein levels for HY5, PIF3 and POR among the genotypes was reproduced in multiple independent experiments, and agreed with our previous data set in the older version of Figure 7. However, when we repeated the quantification of transcripts using the full suite of germplasm, we uncoupled a discrepancy in the expression of PIF3 and HY5 transcript levels. We have tested different conditions that may have affected the qPCR transcript data of etiolated seedlings in our previous submitted version of Figure 7A, and attributed the discrepancy to a variability in the etiolated growth and tissue collection environment, including light leakage sufficient to change gene expression, but not initiate photomorphogenesis. A sentence addressing this has been added to the Discussion (subsection “An apocarotenoid signal regulated PIF3 and HY5 transcript levels”) and additional data sets generated in full darkness (Figure 7) or light (Figure 8) now resolve the above discrepancy.

It is also unclear why, when there are major changes in transcript abundance for PIF3 and HY5 in ccr2, it was concluded so strongly that regulation of the PIF3 and HY5 protein levels was post-transcriptional. The regulation of PIF3 and HY5 transcript abundance in ccr2 with and without D15 needs to be completed before this can be concluded.

Done. We agree with the reviewer’s comments. Please see Figures 7 and 8 for the inclusion of *ccr2* and *det1-154* germplasm and new quantification of POR, PIF3 and HY5 transcript as well as protein levels.

Moreover, it is concluded (subsection “An apocarotenoid post-transcriptionally regulates PIF3 and HY5 protein levels”) that the lack of a PLB in ccr2 is not due to the reduction in PIF3, but other PIFs (especially PIF1) have not been tested and multiple pif mutants do lack PLBs even if pif3 alone doesn't.

We did not test other *pif* mutants since it has been demonstrated that the *pifq* quadruple mutant (*pif1 pif3 pif4 pif5*) contains a PLB, albeit reduced in size compared to WT (See Martin et al., 2016, Figure 1F). This paper is cited accordingly in the Discussion.

The relationship with det1 is quite difficult to understand. det1 mutants lack a PLB and therefore the rescue of PLBs in ccr2 det1-154 is extremely surprising in my view. A rescue in nuclear gene expression might account for it in part, but, although expected, this is not tested properly. Finally, there is also confusion in the related discussion of POR. Even in the Abstract it is stated that "ccr2 has no PLB and normal POR" as well as that "the apocarotenoid signal and det1 complemented each other to restore POR levels and PLB formation". If the ccr2 phenotype is mediated by the production of apocarotenoids it is hard to reconcile these two statements.

Corrected. We agree and thank you for finding this error. The sentence has been corrected as follows: "The apocarotenoid signal restored POR protein levels and PLB formation in *det1*, thereby controlling plastid development ".

The statement in the second paragraph of the subsection “An apocarotenoid signal regulates skotomorphogenesis and plastid biogenesis in parallel to DET1” also cannot be true as it seems quite clear that the level of POR is not controlling PLB formation in this system (as opposed to the experiments of Armstrong et al. in which POR levels were modulated).

Corrected. We agree with the reviewer’s comment and have removed “to control PLB formation” to improve accuracy of statement.

The explanation later in the aforementioned subsection for det1 rescue of ccr2 doesn't make sense to me either. If the cis-carotene cleavage product is up-regulating POR and enhancing Pchlide levels (no evidence of this) to enable PLB formation why does ccr2 not have a PLB? The det1 mutant would only counter the POR effect.

Corrected. Thank you for noting the errors in our sentence structure that led to confusion. Sentence formation and wording has been corrected to improve clarity.

In summary, this is a fascinating system that has the potential to reveal some important regulatory pathways in the control of plastid development, but some of the logical inconsistencies highlighted above need to be ironed out to make this story more coherent to the reader.Reviewer #2:[…] 1) The visible phenotype of CRITSO mutants, its basis and its relevance. This phenotype is variously described as "variegated" or "virescent", and is referred to as triggered by short days, a potential seasonal signal. This matters because the terminology suggests both mechanistic basis and fitness significance. A variegated mutation causes a stochastic failure in chloroplast development in some young, differentiating cells, but not others. As a result, some differentiated cell clusters/regions will show complete failure of greening, and others normal greening. This does not describe the phenotype of ccr2, where all young cells show impaired greening, and all completely differentiated ones appear normal. This is virescence. a kind of delayed chloroplast development. The rice CRITSO mutant, zebra2, shows transverse pale and green stripes, while variegated cereal mutants (like barley albostrians) show longitudinal stripes. The study of Han et al., 2012, showed that stripes of zebra2 disappeared when plants were grown under continuous light, even though the carotenoid composition appeared unaffected by photoperiod. In other words, the pale stripes resulted from periods in which chloroplast differentiation in leaf meristematic cells occurred in the dark. This study is cited but such observations are not. The observations with Arabidopsis ccr2 are uncannily similar, the short photoperiod causing virescence being, more importantly, a long daily dark period. The most parsimonious explanation for the phenotype is the occurrence of a phenomenon or accumulation of a product in the dark, one which, upon light exposure, causes chloroplast development impairment, deceleration (arabidopsis and tomato) or arrest (rice).

Corrected. Thank you for this important clarification and oversight. We agree the phenotype reflects virescence and not variegation. We have made changes throughout the manuscript to reflect the above and below comments.

2) The consequences of the action of ACS in the dark were described by Han et al. as production of reactive oxygen species, particularly singlet oxygen, upon light exposure of accumulated tetra-cis-lycopene. The authors of the current manuscript rightly highlight the differences between the transcriptome programme of ccr2 seedlings or virescent leaves and those related to the "genomes uncoupled" regulation of photosynthesis associated nuclear genes. It has been shown (Page et al., 2017) that the repression of such genes by the "FR block of greening" treatment, which in fact impairs the correct etioplast-to-chloroplast transition, is associated to singlet oxygen damage. The source of the damaging oxygen species in this case is accumulated protochlorophyllide in the absence of its reductase, PORA (and of a prolamellar body). A comparison by the authors of the programme in ccr2 and Page data could be revealing as to the root of the ccr2 phenotype.

Thank you. We have re-analysed our dataset in parallel with the Page et al., 2017. We compared DE genes miss-regulated by Far Red (Fr) light and norflurazon (NF) treatment in de-etiolated seedlings from Page et al., 2017 with DE genes miss-regulated in *ccr2* and *ccr2* ziso-155. There was no significant overlap in DE genes miss-regulated by Fr (see Supplementary file 3). This supports our observation that singlet oxygen damage is unlikely to be the root of the *ccr2* generated signal, and further strengthens are observations that a *cis*-carotene cleavage product is the signal that impaired PLB formation in *ccr2*, and the etioplast-to-chloroplast transition following de-etiolation independent of singlet oxygen/GUN signalling pathways. We have highlighted this data in the Results and Discussion. During this analysis, we confirmed an overlap in miss-regulated genes between NF treatment and *ccr2*, that in our first version was reported in Supplementary Table 6, We updated this this table and moved to main paper as Table 2.

3) Particularly if the above turns out to be correct, one may not need to invoke a change in apocarotenoids as the cause for the reversion of ccr2 by det1. Instead, det1 would prevent the occurrence of a skotomorphogenic programme during the daily, extended dark period. Consequently, no photodynamic reagent (unbound protochlorophyllide?), source of singlet oxygen, or otherwise chloroplast repressive signal would accumulate during such periods.

We appreciate your thinking and suggestions for future work. As you note, the amount of data in the current manuscript is already “extraordinary” and we will address this line of experimentation in future investigations.

4) The more parsimonious interpretation above does not diminish the importance of the ACS in the development of etioplasts or pre-chloroplasts. If could, instead, throw a new angle on it and emphasise its importance. It appears that the modified skotomorphogenic and the virescent stages of ccr2 show commonalities in transcriptome programmes, and in the ratio of key transcription factor drivers of skoto- or photo-morphogenic stages (high and low PIF3/HY5 ratios respectively). As the authors note, this could place the still unidentified ACS signal in a central role in skotomorphogenesis.

Action. The additional data requested by the reviewers has enabled us to incorporate the reviewer’s advice into our modified models.

5) Some of the protein blots in Figure 7 could have had separate control genotypes or conditions. Some description of exposures would help. See below.

Corrected. Thank you for your advice. We have fixed the problems in the new Figures 7 and 8.

6) The entire manuscript text uses a degree of confidence in the enzymatic activities and their responsible genes which the reviewer, admittedly not a specialist, wonders whether it can be justified. Early carotenoid biosynthesis genes are strongly seedling lethal. Losses of many later genes cause profile differences but relatively healthy development, which would be consistent with a combination of enzyme- and non-enzyme-mediated activities, or with redundancy in gene function, or both. This is referred in relation to the phenotypes of individual ccd mutants, but one wonders whether similar caution should be used throughout.

Thanks, we appreciate your comments about not overstating the complexity of functional redundancy with enzymatic and non-enzymatic activities and have ensured our language is consistent with best practice and understanding in the carotenoid community. We have revised accordingly to insure caution in our interpretation.

[Editors' note: further revisions were suggested prior to acceptance, as described below.]

The manuscript has been improved but there are some remaining issues that need to be addressed before acceptance, as outlined below:The reviewers request more precision in describing "FR-block of greening". During the review consultation, the reviewers also discussed that the term "leaf meristematic cells" could be misleading and that the term "differentiating cells" would be more suitable.

We thank the reviewers for their constructive suggestions and valuable comments. We have addressed all issues as suggested. In summary, we corrected: 1) the descriptions related to data interpretation from Page et al., 2017; 2) terminology related to cells in the leaf primoridia; and 3) added a Figure 7—figure supplement 1 gel blot to show the reduction in DET1-154 peptide size compared to DET1.

Reviewer #2:[…] The authors, in this reviewer's reading, describe that study of Page et al., and the impact of their experimental intervention, incorrectly. This is important because it obscures the relevance of the comparison and the importance of the findings. The so-called "FR-block of greening" cannot be described as "the action of FR light" or "a FR treatment". Rather it is the action of WHITE light on seedlings which had PREVIOUSLY been deetiolated under FR light. Such pretreatment results in a repression of PORA activity, while synthesis of Protochlorophyllide continues without a concomitant conversion into chlorophyllide. As a consequence, subsequent exposure to white light leads to absorption by Pchlide and generation of singlet oxygen, with lethal consequences for the seedlings overall. It was not impossible to conceive before that the extended dark periods resulted in the establishment of a similar condition in the ccr2 mutant. The authors have now both ruled this out and provided an alternative explanation for the "corrective" effect of reducing dark periods. This issue, addressed at multiple points in the manuscript's text, needs redressing.

Fixed at all multiple points.